

# Postcranial anatomy and histology of *Seymouria,* and the terrestriality of seymouriamorphs

Kayla D. Bazzana[1,2], Bryan M. Gee[1], Joseph J. Bevitt[3] and Robert R. Reisz[1,4]

[1] Department of Biology, University of Toronto Mississauga, Mississauga, Ontario, Canada
[2] Department of Natural History, Royal Ontario Museum, Toronto, Ontario, Canada
[3] Australian Centre for Neutron Scattering, Australian Nuclear Science and Technology Organisation, Lucas Heights, New South Whales, Australia
[4] International Center of Future Science, Dinosaur Evolution Research Center, Jilin University, Changchun, Jilin Province, China

Corresponding author
Kayla D. Bazzana,
kayla.bazzana@mail.utoronto.ca

## ABSTRACT

*Seymouria* is the best known of the seymouriamorphs, a group of Permo-Carboniferous reptiliomorphs with both terrestrial and aquatic taxa. The majority of research on *Seymouria* has focused on cranial anatomy, with few detailed descriptions or illustrations of the postcrania. We utilized neutron computed tomography (nCT) and histological sampling to provide updated, detailed figures that clarify details of the postcranial anatomy and to assess the development and histology of *Seymouria* through specimens from the early Permian Richards Spur locality. The correlation of morphological and histological data indicate rapid metamorphosis in this terrestrially capable stem amniote, with the youngest specimen being postmetamorphic despite being distinctly younger than premetamorphic individuals of *Discosauriscus*, the only other seymouriamorph to have been histologically sampled. The microanatomical data (e.g., semi-open medullary cavity) also substantiate the hypothesis that *Seymouria* was terrestrial based on interpretation of external features, although the persistence of a modestly developed medullary spongiosa in comparison to either *Discosauriscus* or to other co-occurring terrestrial tetrapods suggests additional nuances that require further exploration. In the absence of clearly recognizable postmetamorphic stages in several seymouriamorph taxa, it is difficult to determine the evolutionary trajectory of terrestriality within the clade. Our analysis provides the first histological characterization of the life history of *Seymouria* and highlights the need for further study of seymouriamorph ontogeny.

## INTRODUCTION

Seymouriamorphs are among the best-known stem amniotes (but see Marjanović & Laurin, 2019 for alternative phylogenetic placement) and are known primarily from Lower Permian deposits in North America, Europe, and Russia (*Broili, 1904*; *Amalitzky, 1921*; *White, 1939*; *Vaughn, 1966*; *Berman, Reisz & Eberth, 1987*; *Berman & Martens, 1993*; *Sullivan & Reisz, 1999*; *Bulanov, 2014*). Within seymouriamorphs, only the monotypic family Seymouriidae

is known from both North America and Eurasia (*Broili, 1904*; *White, 1939*; *Vaughn, 1966*; *Berman & Martens, 1993*). Of the North American localities, the materials of *Seymouria* with the greatest detail of preservation are known from the Dolese Brothers Limestone Quarry near Richards Spur, Oklahoma (*Sullivan & Reisz, 1999*; *Bazzana et al., 2020*). However, the only previously described postcranial material from this locality consists of a few isolated elements (*Sullivan & Reisz, 1999*). Furthermore, the most detailed description and figuring of the postcrania of *Seymouria* from any locality is that completed by *White (1939)*. Subsequent authors have provided focused descriptions of the atlas-axis complex (*Berman, Reisz & Eberth, 1987*; *Sumida, Lombard & Berman, 1992*) and the manus and pes (*Berman et al., 2000*), but *White*'s (*1939*) work remains the most thorough description of the postcranial skeleton in its entirety. While the interpretations made by *White (1939)* have been largely supported, or at least not overturned, by subsequent authors, his illustrations were unlabelled, somewhat stylized, and from slightly angled, non-standard perspectives, which collectively limits their utility. As stem amniotes that are well-documented in the fossil record, seymouriamorphs provide a relatively accessible window through which to examine morphological changes associated with terrestrial lifestyles in Paleozoic tetrapods, and updated osteology of the postcranial skeleton with detailed figures and descriptions that can be readily utilized by other workers is of great import.

Here we describe new postcranial material of *Seymouria* from the early Permian Richards Spur locality, including several articulated vertebrae and a complete humerus, femora, and fibula, and provide updated descriptions, illustrations, and images of the postcrania. Analysis of several limb elements and vertebrae using neutron computed tomography (nCT) and histological sampling provides important details regarding the development and internal anatomy of the postcrania of *Seymouria* and contributes to our understanding of the extent to which these stem amniotes were adapted to terrestrial lifestyles.

## MATERIALS & METHODS

### Neutron tomography

Neutron tomography measurements were performed at the DINGO thermal-neutron radiography/tomography/imaging station (*Garbe et al., 2015*) located at, and tangentially facing, the 20 MW Open-Pool Australian Lightwater (OPAL) reactor housed at the Australian Nuclear Science and Technology Organisation (ANSTO), Lucas Heights, New South Wales, Australia. The DINGO facility utilises a quasi-parallel collimated beam of thermal neutrons.

For the femur (ROMVP 80915) and the fibula (ROMVP 80917), the instrument was equipped with an Iris 15TM Large Field of View sCMOS camera (5,056 $\times$ 2,968 pixel, 16-bit) and Zeiss Ikon 100 mm f/2.0 Makro Planar lens. Based on a desired spatial resolution of $\sim$60 $\mu$m across the partially-embedded femur and fibula, a maximum sample width of 48.5 mm and minimum sample-to-detector distance of 28 mm, the DINGO instrument was configured with a 30 $\mu$m thick terbium-doped Gadox scintillator screen (Gd2O2S:Tb, RC Tritec AG) and 25.2 $\times$ 25.2 $\times$ 25.2 $\mu$m voxels for a Field-of-View of 100 $\times$ 74.5 mm.. To maximise counting statistics and minimise subsequent noise in the

tomographic reconstruction, a collimation ratio (*L/D*) of 500 was used, where *L* is the neutron aperture-to-sample length and *D* is the neutron aperture diameter. This high-flux configuration traditionally illuminates a 200 mm ×200 mm area around the sample area with 4.75 × 107 neutrons cm-2s-1, leading to high background radiation and zingers on the detector. A newly installed slit system was implemented to restrict the neutron-irradiated area about the specimen to achieve optimum scan conditions and a divergence-limited spatial resolution of 56 µm. A total of 900 equally-spaced angle shadow-radiographs were obtained every 0.20° as the sample was rotated 180° about its vertical axis. Both dark (closed shutter) and beam profile (open shutter) images were obtained for calibration before initiating shadow-radiograph acquisition. To reduce anomalous noise, a total of three individual radiographs with an exposure length of 4.0 s were acquired at each angle (*Mays, Bevitt & Stilwell, 2017*) for a total scan time of 4.6 h.

For the sacral series (OMNH 79348), a collimation ratio (*L/D*) of 1,000 (*Garbe et al., 2015*) was used to ensure highest available spatial resolution. The field of view was set to 200 × 200 mm$^2$ with a voxel size of 73.3 × 73.3 × 73.3 µm and sample to detector distance of 70 mm. Neutrons were converted to photons with a 100 µm thick ZnS(Ag)/$^6$LiF scintillation screen (RC Tritec AG); photons were then detected by an Iris 15 sCMOS camera (16-bit, 5,056× 2,960 pixels) coupled with a Makro Planar 24 mm Carl Zeiss lens. The tomographic scan consisted of a total of 720 equally-spaced angle shadow-radiographs obtained every 0.25° as the sample was rotated 180° about its vertical axis. Both dark (closed shutter) and beam profile (open shutter) images were obtained for calibration before initiating shadow-radiograph acquisition. To reduce anomalous noise, a total of three individual radiographs with an exposure length of 8 s were acquired at each angle (*Mays, Bevitt & Stilwell, 2017*). These individual radiographs were summed in post-acquisition processing using the Grouped ZProjector function in ImageJ v.1.51 h. Total scan time was 5.7 h.

The individual radiographs were summed in post-acquisition processing using the 'Grouped ZProjector' plugin in ImageJ v.1.51 h in accordance with our previous measurements; this plugin was developed by *Holly (2004)*. Tomographic reconstruction of the 16-bit raw data was performed using commercially available Octopus Reconstruction v.8.8 software package and the filtered back-projection algorithm to yield virtual slices perpendicular to the rotation axis. When these slices are stacked in a sequence, they form a three-dimensional volume image of the sample. The reconstructed volume data were downsampled by a factor of 2 in ImageJ to reduce computation time, then rendered and segmented with Avizo Lite 9.3.0.

### Histology

Histological preparation followed standard procedures (*Padian & Lamm, 2013*). All specimens were photographed prior to embedding in EP4101UV resin (Eager Polymers), which was allowed to cure for 24 h. ROMVP 80916 (partial femur), ROMVP 81198 (vertebra), and ROMVP 81199 (vertebra) were prepared at the Royal Ontario Museum (ROMVP), Toronto, Canada. Specimens were cut on the IsoMet 1000 precision saw (Buehler) and mounted to frosted plexiglass slides with cyanoacrylate adhesive. For the femora, the cut was made at the approximate region of the minimum diaphyseal

circumference; for the vertebra, the first cut was made sagittally (anteroposteriorly) down the midline, and the second cut was made transversely through one of the two halves of the block. For ROMVP 80916 (larger, partial femur), the section is taken slightly proximal to the inferred minimum circumference due to the incomplete specimen's nature.

Mounted blocks were trimmed using the IsoMet and ground on the Hillquist Thin Sectioning Machine lapidary wheel. Manual polishing using 1,000-mesh grit on glass plates and a combination of 1-μm and 5-μm grit on polishing cloths was used to remove scratches. ROMVP 81200 (partial femur) was prepared in a similar fashion but with a different equipment setup at the University of Toronto Mississauga. Cutting was performed on the Metcut-5 low speed saw (MetLab), initial grinding on the Metcut-10 Geo (MetLab), and manual grinding on a cutlery whetstone block. Imaging was done on two Nikon AZ-100 microscopes, both fitted with a DS-Fi1 camera and NIS Elements-Basic Research software registered to David C. Evans and to Robert R. Reisz.

Preparation of the specimens was performed by Diane Scott and Bryan M. Gee using pin vises and air scribes. Figures were prepared using Adobe Illustrator and Photoshop. The raw data are reposited online through MorphoBank (*O'Leary & Kaufman, 2012*; see Data Availability statement for additional details).

## SYSTEMATIC PALEONTOLOGY

Order Seymouriamorpha Watson, 1917
Family Seymouriidae *Williston, 1911*
Genus *Seymouria Broili, 1904*

**Horizon and locality.** Dolese Brothers Limestone Quarry near Richards Spur, Oklahoma, USA. Early Permian (Artinskian).

**Referred specimens.** OMNH 74721, right humerus; OMNH 79346, complete vertebra; OMNH 79347, string of 11 articulated vertebrae; OMNH 79348, string of three articulated vertebrae; OMNH 79349, complete vertebra; OMNH 79350, partial vertebra; OMNH 79351, complete vertebra; OMNH 79352, partial vertebra; OMNH 79353, partial vertebra; ROMVP 81198, complete vertebra; ROMVP 81199, complete vertebra; ROMVP 80915, left femur; ROMVP 80916, proximal left femur; ROMVP 80917, left fibula; ROMVP 81200, distal left femur.

## ANATOMICAL DESCRIPTION

**Vertebra.** OMNH 79346, OMNH 79349, OMNH 79350, OMNH 79353, ROMVP 81198, and ROMVP 81199 are isolated presacral vertebrae between the tenth and twenty-second positions (Fig. 1). OMNH 79346 and OMNH 79349 are mostly complete, whereas OMNH 79350 and OMNH 79353 are incomplete. OMNH 79351 and OMNH 79352 are complete vertebrae from the third to ninth vertebral positions (Fig. 2). OMNH 79347 consists of 11 articulated presacral vertebrae, likely between the tenth and twenty-second positions

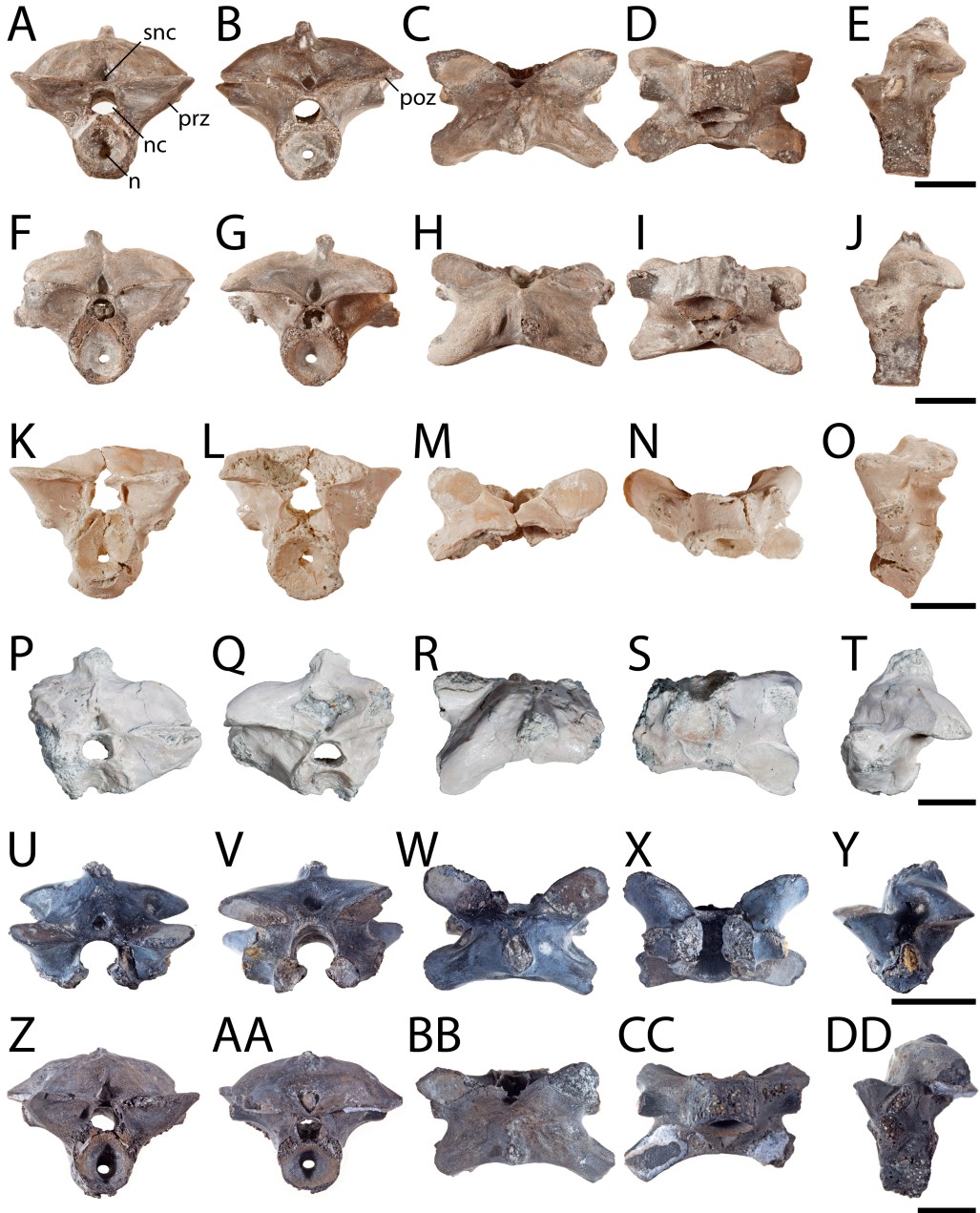

**Figure 1 Posterior presacral vertebrae of *Seymouria*.** (A) OMNH 79346 in anterior view; (B) the same in posterior view; (C), the same in dorsal view; (D) the same in ventral view; (E) the same in left lateral view; (F) OMNH 79349 in anterior view; (G) the same in posterior view; (H) the same in dorsal view; (I) the same in ventral view; (J) the same in left lateral view; (K) OMNH 79350 in anterior view; (L) the same in posterior view; (M) the same in dorsal view; (N) the same in ventral view; (O) the same in left lateral view; (P) OMNH 79353 in anterior view; (Q) the same in posterior view; (R) the same in dorsal view; (S) the same in ventral view; (T) the same in left lateral view; (U) ROMVP 81198 in anterior view; (V) the same in posterior view; (W) the same in dorsal view; (X) the same in ventral view; (Y) the same in left lateral view; (Z) ROMVP 81199 in anterior view; (AA) the same in posterior view; (BB) the same in dorsal view; (CC) the same in ventral view; (DD) the same in left lateral view. Scale bar equals one cm. **n**, notochordal canal; **nc**, neural canal; **poz**, postzygapophysis; **prz**, prezygapophysis; **snc**, supraneural canal.

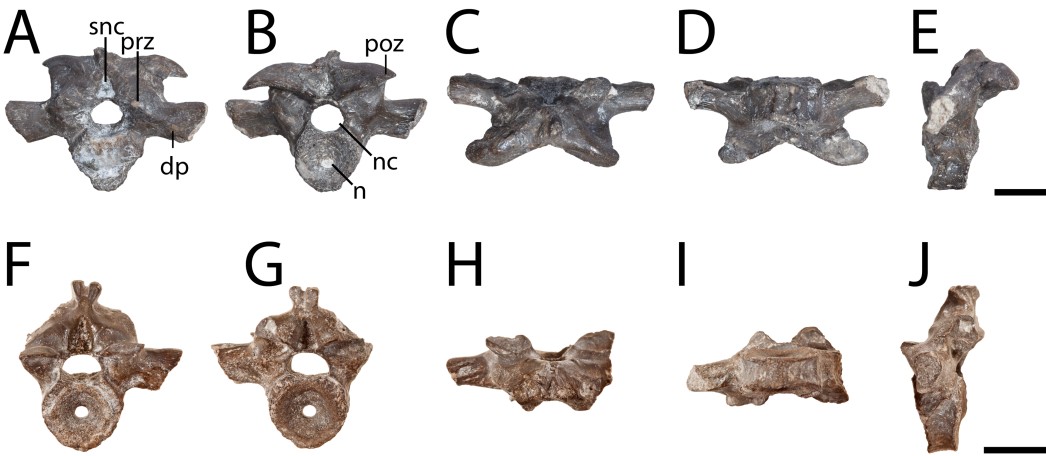

**Figure 2** **Anterior presacral vertebrae of *Seymouria*.** (A) OMNH 79351 in anterior view; (B) the same in posterior view; (C) the same in dorsal view; (D) the same in ventral view; (E) the same in left lateral view; (F) OMNH 79352 in anterior view; (G) the same in posterior view; (H) the same in dorsal view; (I) the same in ventral view; (J) the same in left lateral view. Scale bar equals one cm. **dp**, diapophysis; **n**, notochordal canal; **nc**, neural canal; **poz**, postzygapophysis; **prz**, prezygapophysis; **snc**, supraneural canal.

(Fig. 3A). OMNH 79348 consists of the last presacral position, the first sacral vertebra, and a possible second sacral vertebra in articulation (Figs. 3B–3E). Despite the isolated nature of the specimens, the position of the vertebrae can be inferred based on the morphology of the diapophyses, which change in length, cross-sectional profile, and anteroposterior orientation along the vertebral column (*White, 1939*: Fig. 13).

All the vertebrae described here exhibit the swollen pre- and postzygapophyses that are characteristic of *Seymouria* (*White, 1939*; *Holmes, 1989*). The first sacral vertebra in OMNH 79348 captures the transition from the expanded zygapophyses of the trunk series to the more transversely constricted morphology characteristic of the caudal series (*White, 1939*), with the prezygapophyses being slightly more than twice the width of the postzygapophyses (Fig. 3B). The neural spine of OMNH 79349 is strongly deflected to the right, as has been noted in other specimens of *Seymouria* (*Sullivan & Reisz, 1999*). Both OMNH 79351 and OMNH 79352 display the broadly expanded diapophyses characteristic of the anteriormost presacral vertebrae in *Seymouria* (Fig. 2). In OMNH 79352, the neural spine is bifurcated, with both projections directed dorsolaterally; the neural spines of the other specimens are either broken or missing. *Seymouria* has been previously described as having an irregular distribution of bifurcated and non-bifurcated spines along the presacral region between the tenth position and the twenty-second position (*White, 1939*). OMNH 79352 appears to be the first record of bifurcation of the neural spine in a vertebra from the anteriormost trunk. Although the seemingly complete co-ossification of the neural arch and centrum suggests skeletal maturity, the maturity of these vertebrae cannot be more definitively established given the known challenges of applying skeletochronology to vertebral elements (*Danto et al., 2016*).

**Humerus.** OMNH 74721 is a complete right humerus, exhibiting the tetrahedral shape typical of many reptiliomorphs (Fig. 4). The overall morphology matches that described

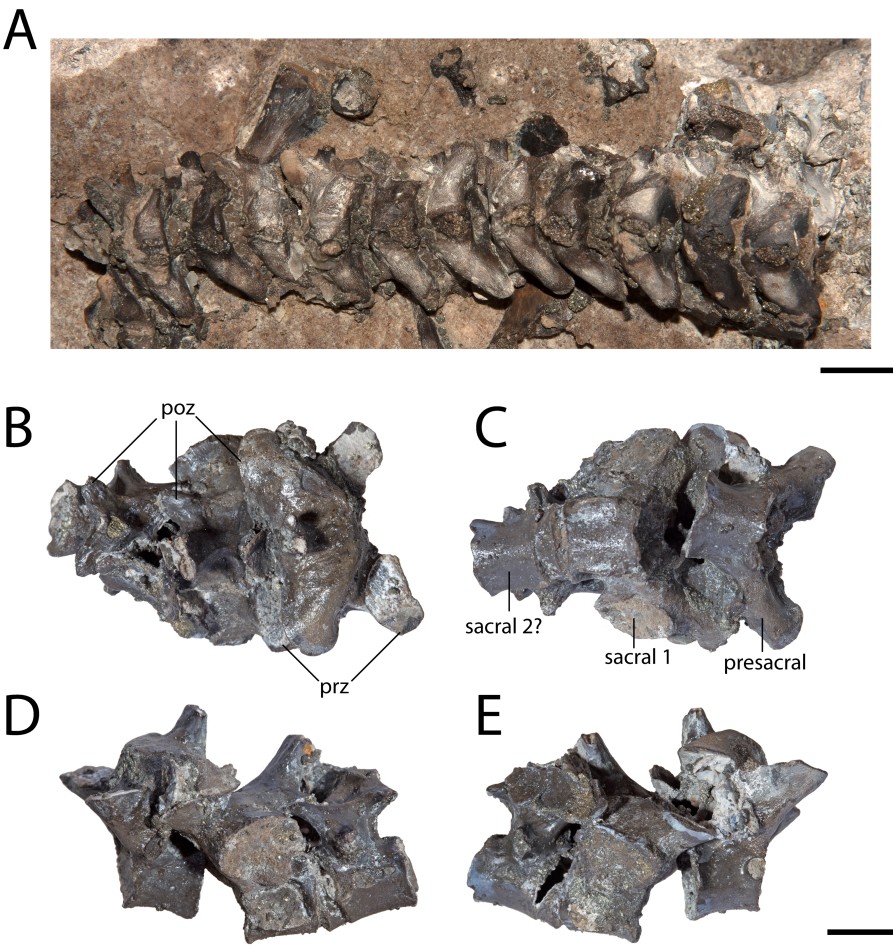

**Figure 3** **Articulated vertebrae of *Seymouria*.** (A) OMNH 79347 in dorsal view; (B) OMNH 79348 in dorsal view; (C) the same in ventral view; (D) the same in left lateral view; (E) the same in right lateral view. Scale bar equals one cm. **poz**, postzygapophysis; **prz**, prezygapophysis.

by previous authors (*White, 1939*; *Sullivan & Reisz, 1999*) in being short and robust with the deltopectoral crest following an L-shaped path. The proximal and distal ends are broadly expanded and are set at an approximately 45-degree angle to each other with no distinct shaft separating the epiphyses. The supinator process is oval in cross-section and located just distal to the short arm of the deltopectoral crest (Fig. 4C). OMNH 74721 exhibits a keel that extends along the anteroventral surface from the deltopectoral crest to the entepicondylar foramen (Fig. 4D); a similar crest has been described by *Sullivan & Reisz (1999)* but was not mentioned by *White (1939)*. The entepicondylar ridge runs along the ventral edge and expands proximally to form what may represent the insertion site for the *m. subcoracoscapularis* on the posterior surface of the humerus, near the glenoid articulation (Fig. 4B). Dorsal to this expansion is a pronounced tubercle for the insertion of the latissimus dorsi muscle *m. latissimus dorsi*. The absence of an ectepicondylar foramen, the proximal position of the insertions for the *m. subcoracoscapularis* and the *m. latissimus*

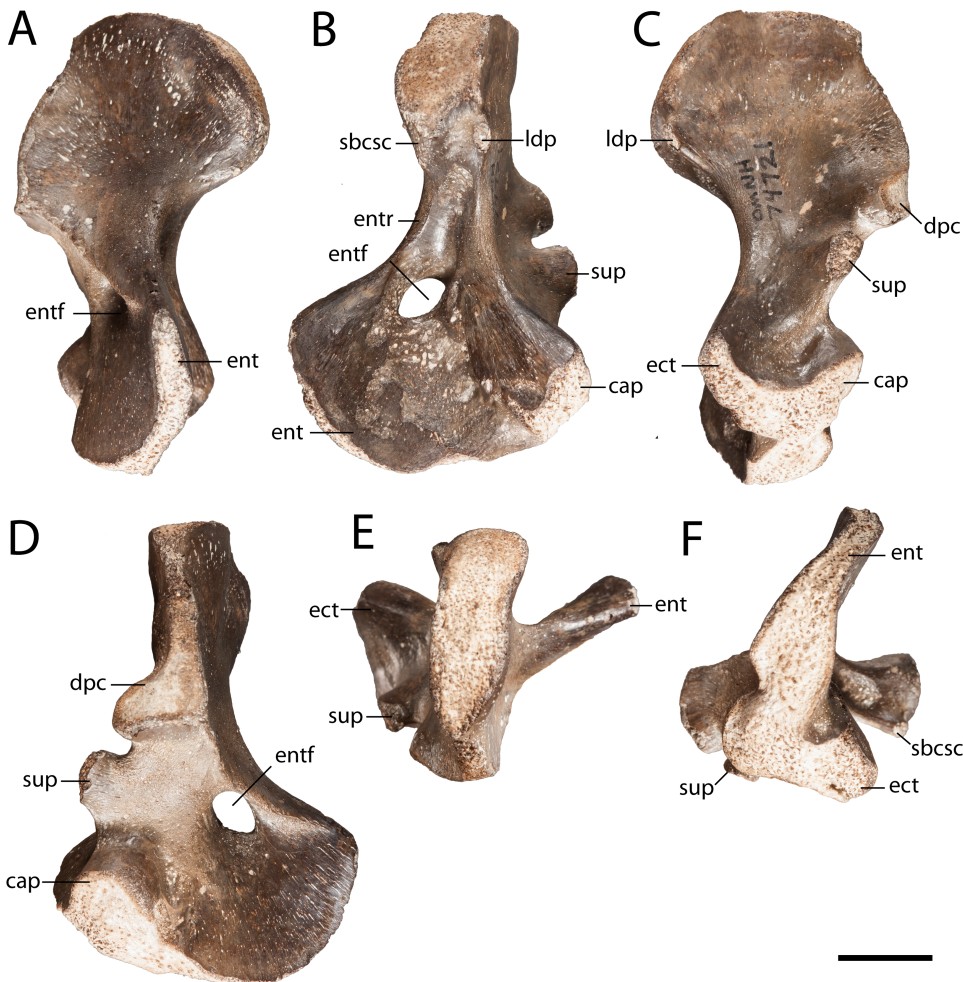

**Figure 4** **OMNH 74721.** (A) anterior view; (B) posterior view; (C) dorsal view; (D) ventral view; (E) proximal view; and (F) distal view. Scale bar equals one cm. **cap**, capitellum; **dpc**, deltopectoral crest; **ect**, ectepicondyle; **ent**, entepicondyle; **entf**, entepicondylar foramen; **entr**, entepicondylar ridge; **sbcsc**, *m. subcoracoscapularis*; **sup**, supinator process.

*dorsi*, and the position and shape of the supinator process all correspond exactly to the descriptions of *Seymouria* given by *White (1939)* and *Sullivan & Reisz (1999)*.

**Femur.** ROMVP 80915 is a complete left femur (Fig. 5), the morphology of which corresponds to previous descriptions (*White, 1939*; *Sullivan & Reisz, 1999*). ROMVP 80916 and ROMVP 81200 are partial left femora (Fig. 6). The element is short and robust. The insertion for the *m. puboischiofemoralis* is visible on the dorsal surface of the head (Figs. 5B, 5D). As is known in *Seymouria* (*White, 1939*; *Sullivan & Reisz, 1999*), the adductor crest extends posterolaterally along the ventral surface of the femur from the trochanter to near the tibial condyle (Fig. 5F). The smaller femur that was previously described by *Sullivan & Reisz (1999*; FMNH PR 2053) was stated to have a prominent trochanter. This appears to remain constant throughout ontogeny, as little difference in the relative size of the trochanter is seen between the smallest (ROMVP 81200) and largest specimens (ROMVP

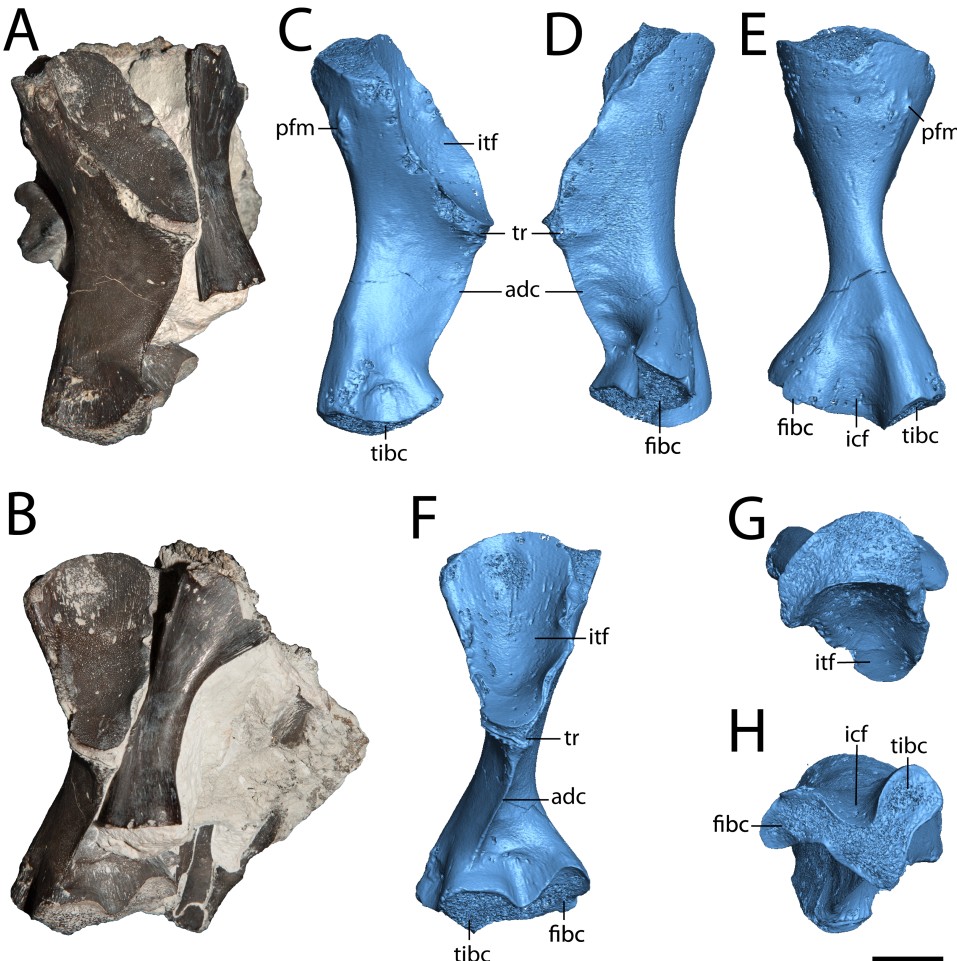

**Figure 5  Hindlimb elements of *Seymouria*.** (A) photograph of ROMVP 80915 and ROMVP 80917 in posterior view; (B) the same in ventral view; (C) digital rendering of ROMVP 80916 in posterior view; (D) the same in anterior view; (E) the same in dorsal view; (F) the same in ventral view; (G) the same in proximal view; (H) the same in distal view. Scale bar equals one cm. **adc**, adductor crest; **fibc**, fibular condyle; **icf**, intercondylar fossa; **itf**, intertrochanteric fossa; **pfm**, insertion for the *m. puboischiofemoralis*; **tibc**, tibial condyle; **tr**, trochanter.

80915 and 80916). The distal articular surface has been described as following an M-shaped curve (*White, 1939*), but this may not be entirely accurate. In ROMVP 80915, ROMVP 81200, and a smaller, more immature femur (FMNH PR 2053; *Sullivan & Reisz, 1999*: Fig. 3), the distal surface is V-shaped, with the tip of the anterior arm being bent ventrally (Fig. 5G), as opposed to the illustration given by *White* (*1939*: Fig. 28) that reconstructs this surface with both ends curving ventrally. However, it is also possible that the shape of the distal surface is ontogenetically variable, as disparities in length suggest that the specimen described by *White (1939)*, measuring 6.4 cm in length, may be more mature than ROMVP 80915 (5.5 cm), the largest complete femur from Richards Spur.

**Fibula.** ROMVP 80917 is a complete left fibula (Fig. 7). Previous descriptions (*White, 1939*; *Sullivan & Reisz, 1999*) have commented only on the general shape of the element, to which

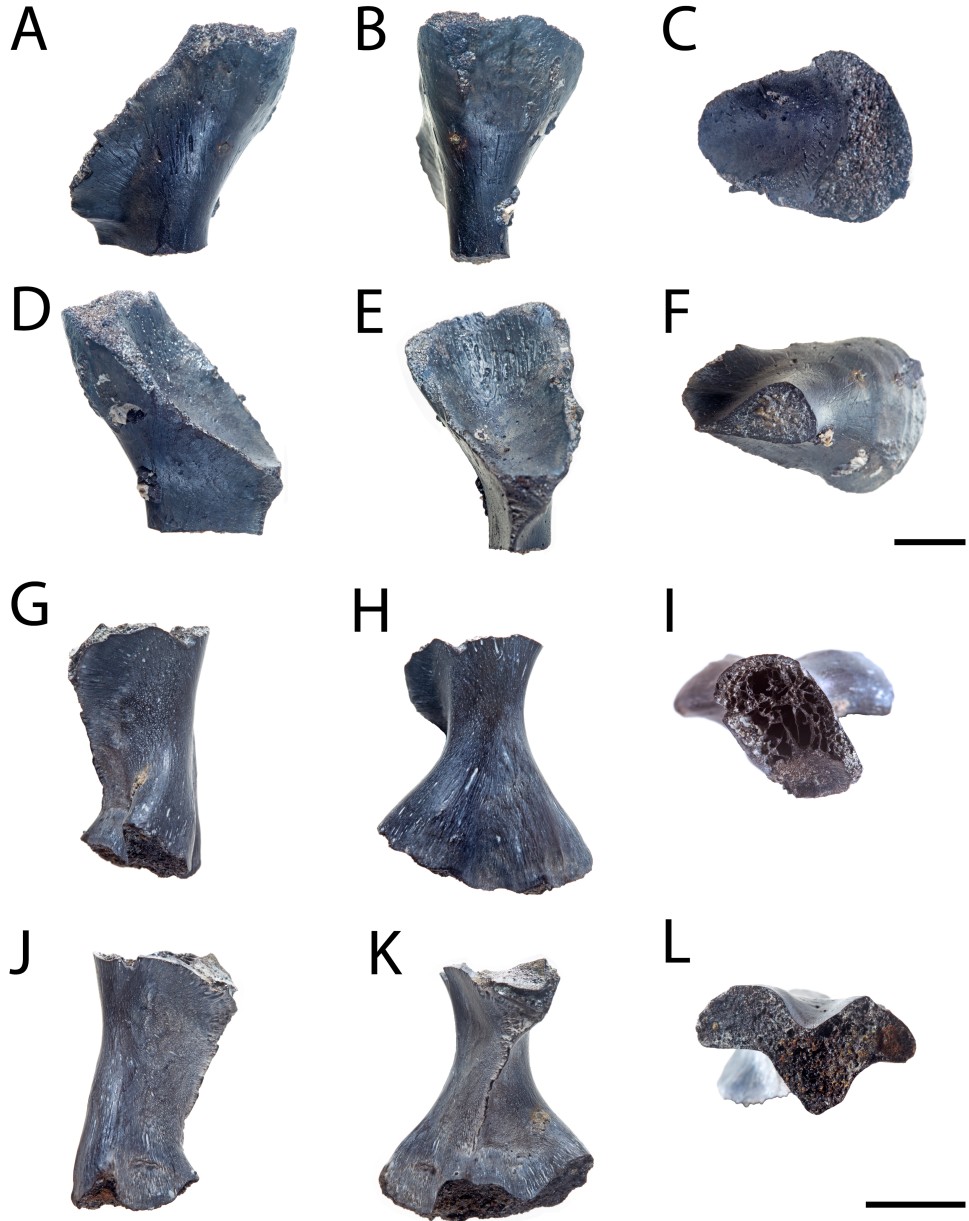

**Figure 6** **Partial femora of *Seymouria*.** (A–F), ROMVP 80916 in anterior, posterior, dorsal, ventral, proximal, and distal views, respectively and (G–L), ROMVP 81200 in anterior, posterior, dorsal, ventral, proximal, and distal views, respectively. Scale bar equals one cm.

ROMVP 80917 corresponds exactly in that the medial surface is deeply concave, the lateral surface is mostly straight with only a slight concavity, and the distal articular surface is more expanded than the proximal surface, which itself is crescentic with its dorsal margin being convex and its ventral margin being concave. Features that have not been described previously in *Seymouria* but that are present in ROMVP 80917 are the pronounced muscle scar along the proximal half of the lateral surface, which may represent the insertion site

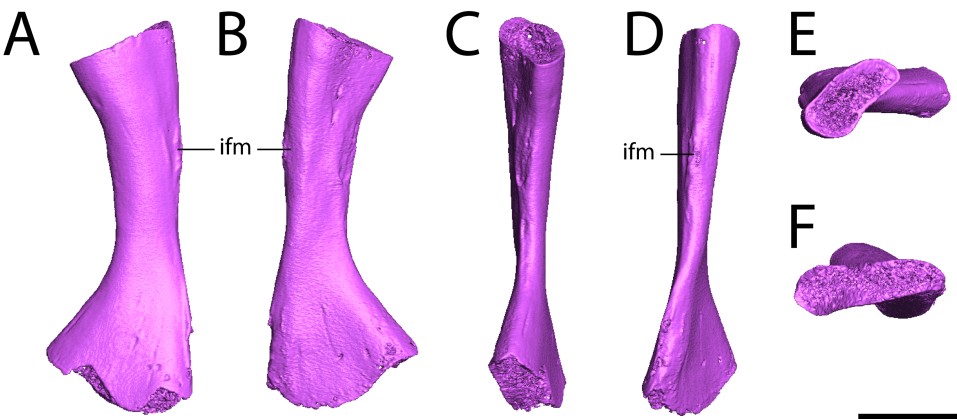

**Figure 7** **Digital renderings of ROMVP 80917.** (A), anterior view; (B) posterior view; (C) medial view; (D) lateral view; (E), proximal view; (F) distal view. Scale bar equals one cm. **ifm**, insertion for the *m. iliofibularis*.

for the *m. iliofibularis* (Figs. 7A–7B, 7D; *Romer, 1956*), and the twisting of the fibular shaft such that the proximal and distal heads lie in planes that are at an approximately 45-degree angle to each other (Figs. 7E–7F).

## HISTOLOGICAL DESCRIPTION

**Vertebra.** ROMVP 81198, an isolated presacral neural arch of a small-bodied individual (Fig. 1E), and ROMVP 81199, a presacral neural arch with centrum of a larger individual (Fig. 1F), were histologically sectioned. The transverse profile reveals a similar microanatomy and histology to that of a previously sampled specimen (OMNH 73499) from Richards Spur (*Danto et al., 2016*). The pleurocentrum of ROMVP 81199 is formed by two domains (Fig. 8), an well-ossified yet porous periosteal domain along the ventrolateral margin of the element that is formed by a thin layer of lamellar bone, and a less dense and more disorganized endochondral domain with trabeculae; this is captured in both sagittal and transverse sections. Although the suture between the pleurocentrum and the neural arch is not clearly defined externally, it is very apparent in the transverse section in which an oblique separation (Fig. 8F) demarcates a largely cartilaginous connection at the time of death. Neither the notochordal canal nor the neural canal were captured in the half-transverse section. The neural arches of ROMVP 81198 and 81199 are very similar in transverse section. Each arch is comprised of a spongy bone texture with a hollow interior. In the smaller ROMVP 81198, the ventral portion of the arch is poorly ossified, with a sparse network of trabeculae (Fig. 9). The sagittal sections are also generally comparable, but a few differences may be noted. Indentations on the anterior and posterior surfaces representing the vestiges of the supraneural canal are present; these indentations are more prominent in the larger ROMVP 81199, and in each specimen, the anterior indentation is more pronounced (Figs. 8A, 9A). In both specimens, the cortex of the neural arch is much thicker along the posterodorsal surface behind the neural spine and extending down to the posterior indentation of the supraneural canal when compared to the anterior surface.

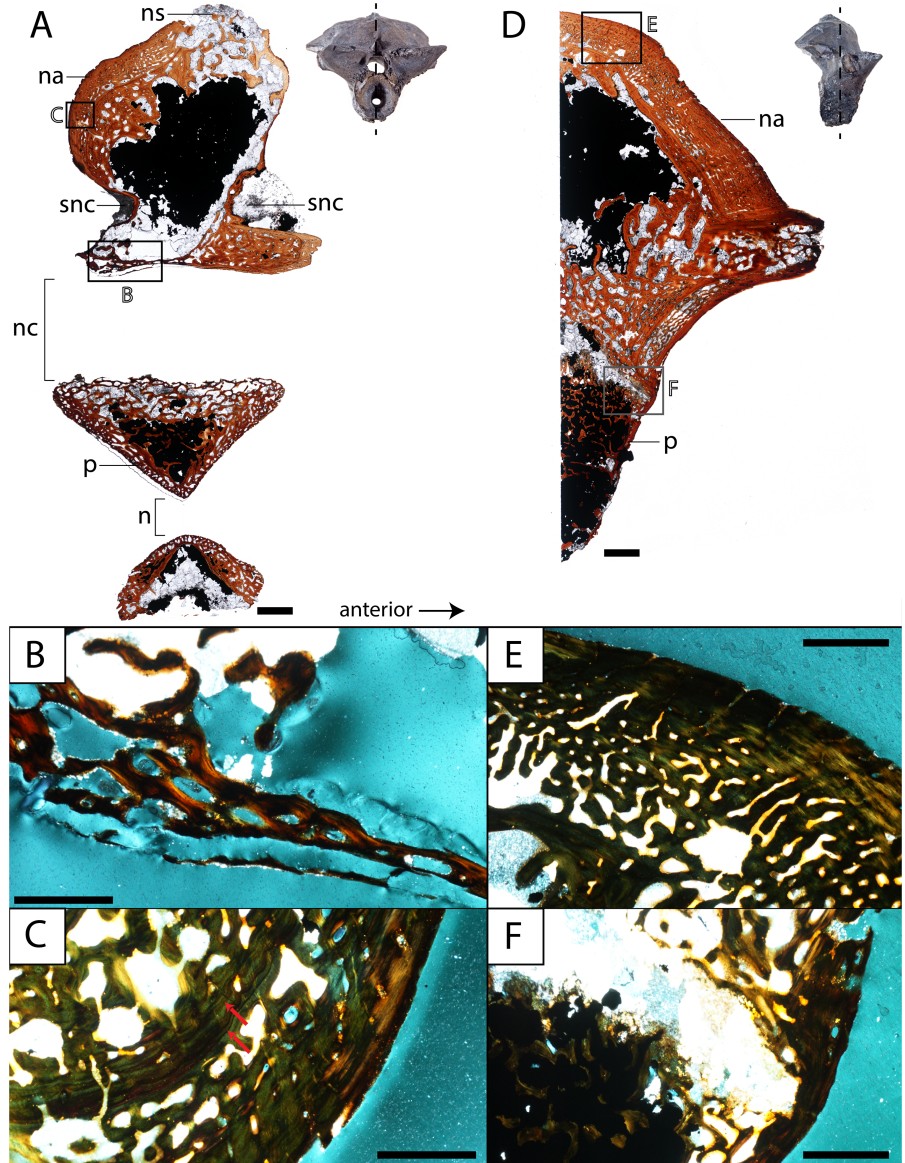

**Figure 8** **Histological sections of a presacral vertebra (ROMVP 81199) of *Seymouria*.** (A) Full sagittal section; (B) close-up under cross-polarized light of the ventral border of the neural arch showing remodelling; (C) close-up under cross-polarized light of the thickened posterior surface; (D) half transverse section; (E) close-up under cross-polarized light of the dorsal surface; (F) close-up under cross-polarized light of the cartilaginous separation between the neural arch and the centrum. Scale bars equal to one mm (A, D); 250 μm (B–C, E–F). **n**, notochordal canal; **na**, neural arch; **nc**, neural canal; **ns**, neural spine; **p**, pleurocentrum; **snc**, supraneural canal.

The thickened region of cortex is far more developed in the larger ROMVP 81199. The neural spines are very poorly ossified. The ventral portion of the arch, which roofs the neural canal, is thin in both specimens and exhibits distinct remodelling (Figs. 8B, 9B). Remodelling is otherwise found mostly near the geometric center of the element (Fig. 9B).

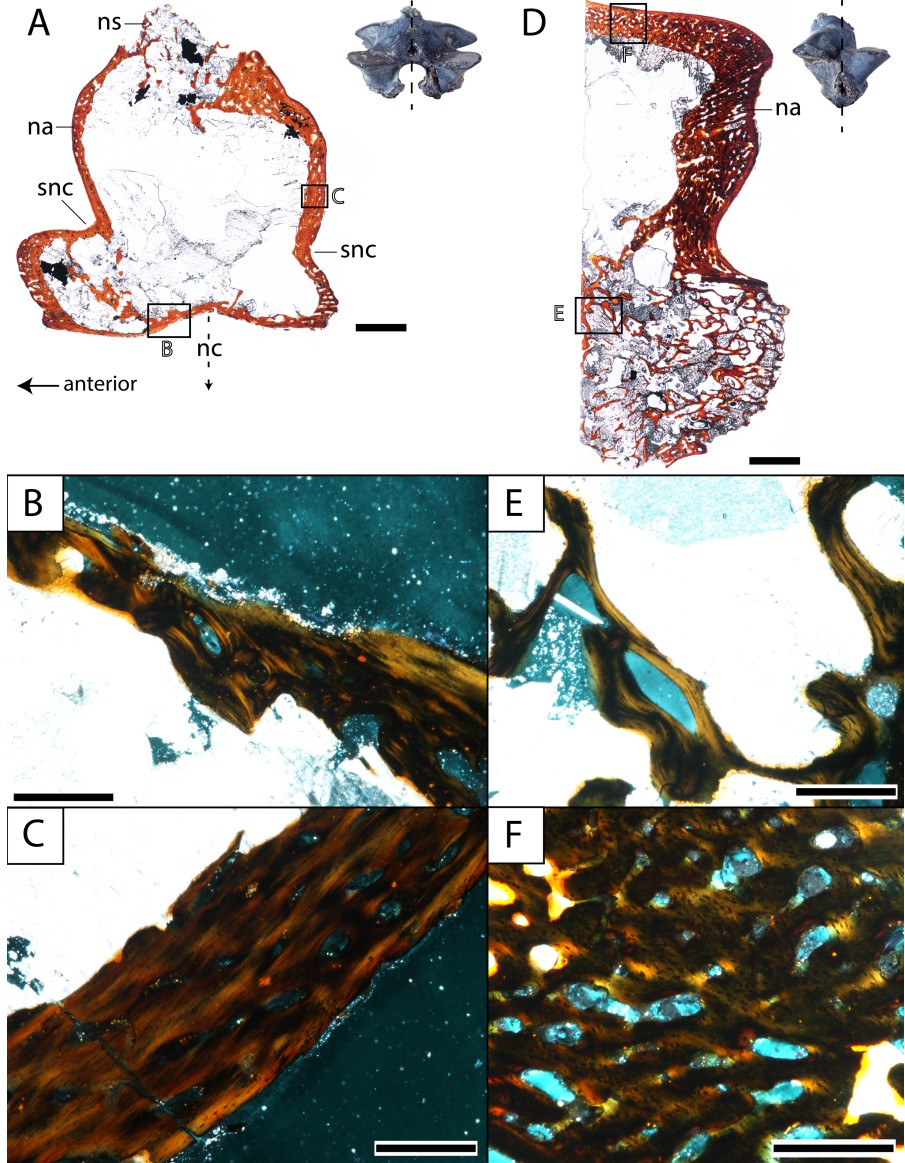

**Figure 9  Histological sections of a presacral vertebra (ROMVP 81198) of *Seymouria*.** (A) Full sagittal section; (B) close-up under cross-polarized light of the ventral border of the neural arch showing remodelling; (C) close-up under cross-polarized light of the thickened posterior surface; (D) half transverse section; (E) close-up under cross-polarized light of the dorsal surface; (F) close-up under cross-polarized light of the geometric center of the neural arch. Scale bars equal to one mm (A, D); 250 µm (B–C, E–F). **na**, neural arch; **nc**, neural canal; **ns**, neural spine; **snc**, supraneural canal.

A pair of closely spaced, distinctive growth lines can be identified in the thickened region of ROMVP 81199 (Fig. 8A).

The three articulated vertebrae of OMNH 79348 were digitally sectioned using the neutron tomography data (Fig. 10). Viewed in transverse section, the data reveal pronounced differences in the compactness of the diapophyses; the diapophyses of the

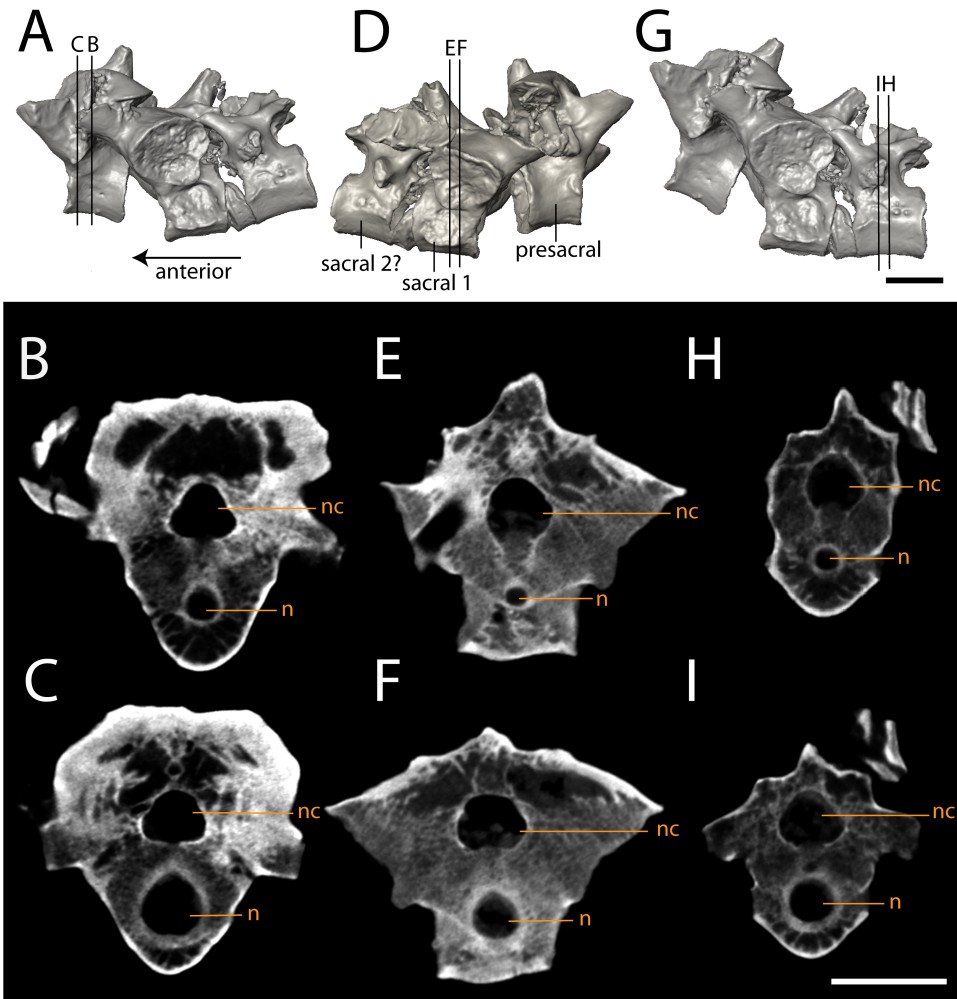

**Figure 10  Isolated profiles of OMNH 79348.** (A, D, G), digital renderings indicating location of digital sections; (B–C), sagittal sections of presacral vertebra at mid-centrum (B) and mid-diapophysis (C); (E–F) sagittal sections of sacral vertebra at mid-centrum (E) and mid-diapophysis (F); (H–G), sagittal sections of caudal vertebra at mid-centrum (H) and mid-diapophysis (I). Scale bar equals one cm. **n**, notochordal canal; **nc**, neural canal.

first sacral vertebra are not only greatly expanded, as is visible externally, but are also substantially less porous than those of either the presacral or the possible second sacral. The scan resolution is not sufficient to permit tissue identification.

**Femur.** ROMVP 80916 and ROMVP 81200 are partial left femora (Fig. 6); ROMVP 80916 is equivalent in size to ROMVP 80915 and is broken distal to the trochanter, while ROMVP 81200 is substantially smaller (about 33% smaller) and is broken proximal to the trochanter. ROMVP 80916 was histologically sectioned slightly proximal to the minimum diaphyseal circumference (Fig. 11), while ROMVP 81200 was sectioned at this minimum region (Fig. 12). ROMVP 80915 was digitally sectioned at the minimum circumference (Fig. 13). All three femora are relatively similar in their microanatomical and histological features. The cortex is relatively compact and well-vascularized, comprised primarily of

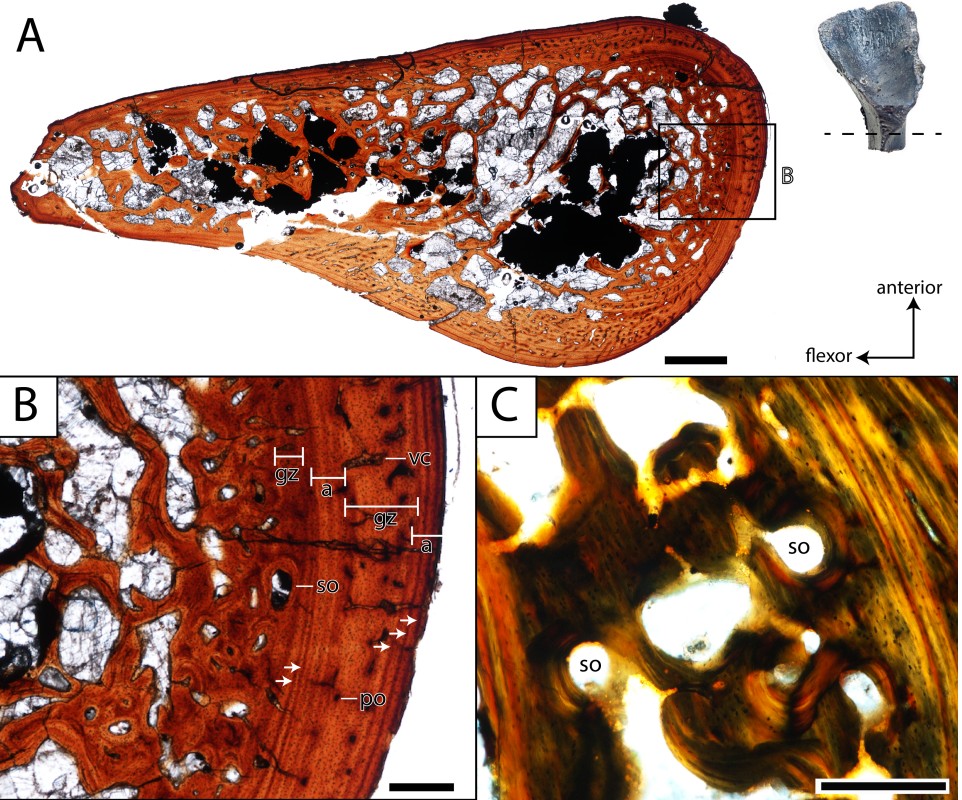

**Figure 11** **Histological section of a partial femur (ROMVP 80916) of** *Seymouria.* (A) full cross-section near the minimum diaphyseal circumference; (B) close-up of the cortical bone; (C) close-up under cross-polarized light of the remodelled bone interface between the cortical bone and the medullary cavity. Note that (C) is from a different thin section than in (A) and thus an inset magnification box is not marked. Scale bars equal to one mm (A); 250 µm (B–C). **a**, annulus; **gz**, growth zone; **po**, primary osteon; **so**, secondary osteon; **vc**, vascular canal. Arrows mark rest lines.

lamellar bone and with a plexiform arrangement of vascular canals. Primary osteons and vascular canals are abundant in the growth zones of both specimens (Figs. 11–12), but ROMVP 81200 appears to have a denser concentration, implying a greater immaturity and a faster growth rate at the time of death. Remodelling is found at the boundary of the medullary cavity in both specimens (Figs. 11C, 12C), but there is distinctly more along the margin of the cavity in the larger ROMVP 80916 (Figs. 11–12). Secondary osteons associated with this remodelling are found in both histologically sectioned specimens. The medullary cavity is relatively open, although there is a network of trabeculae throughout; secondary remodelling is present within this network, but calcified cartilage is absent. The trabecular network appears to be less dense in the digital section of ROMVP 80915 (Fig. 13) than in the histological section of the comparably sized ROMVP 80916 (Fig. 11), but this may be a combination of a slightly more proximal plane of section in the latter and limitations on scanning resolution (25.2 µm) for the former.

In both histologically sectioned specimens, there are numerous circumferential lines; none could be identified in the digital data for ROMVP 80915. In the smaller ROMVP

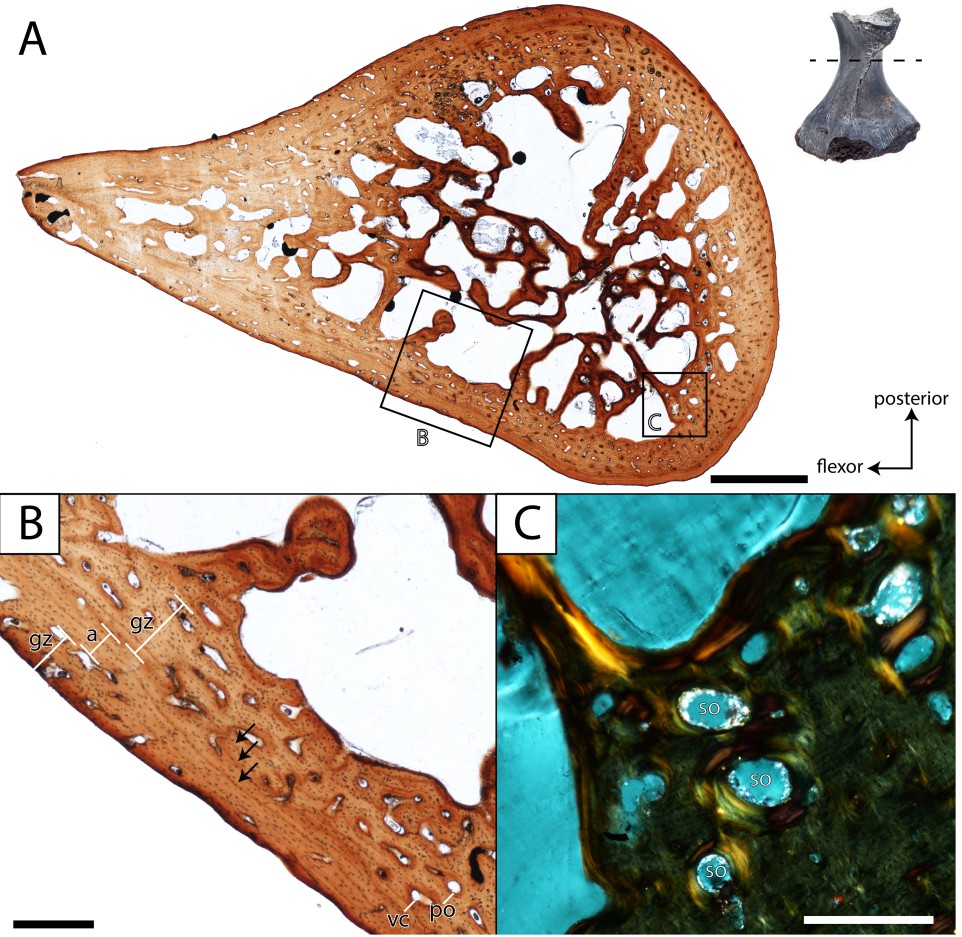

**Figure 12** **Histological section of a partial femur (ROMVP 81200) of *Seymouria*.** (A) full cross-section near the minimum diaphyseal circumference; (B) close-up of the cortical bone showing four lines of arrested growth (LAGs) marked by black arrows; (C) close-up under cross-polarized light of the remodelled bone interface between the cortical bone and the medullary cavity. Scale bars equal to one mm (A); 250 μm (B–C). **a**, annulus; **gz**, growth zone; **po**, primary osteon; **so**, secondary osteon; **vc**, vascular canal. Arrows mark rest lines.

81200, at least three lines are present (Fig. 12B). In the larger ROMVP 80916, at least seven lines are present (Fig. 11B). Based on the distribution of these lines as distinct clusters of closely spaced lines in the poorly vascularized regions, these probably represent multiple rest lines within an annulus (the poorly vascularized region), rather than clusters of lines of arrested growth (LAGs) with highly uneven growth between deposition of each LAG. In ROMVP 80916, there is an annulus partially preserved toward the extensor surface (away from the adductor crest) with at least five rest lines present (Fig. 11A). Most of this annulus has been obliterated by remodelling, and the rest lines are not uniformly distinct around their partial circumference. There are a few vascular canals and primary osteons in this region near the border of the medullary cavity that likely represent a partially preserved growth zone. The partial annulus is followed by a thicker region of well-vascularized tissue

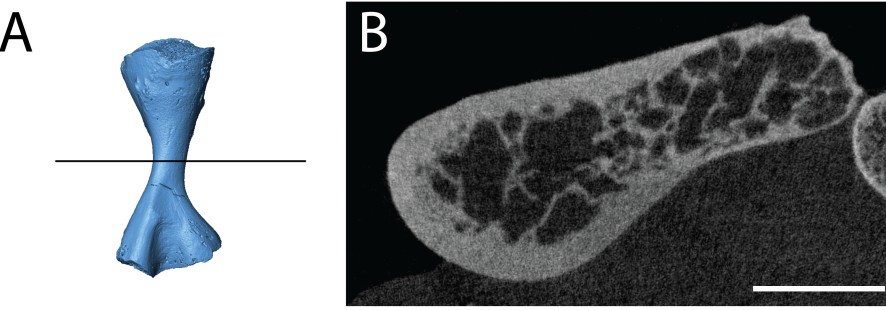

**Figure 13  Digital section of a femur (ROMVP 80915) of *Seymouria*.** (A) digital rendering showing location of section; (B) transverse digital section. Scale bar equals five mm.

(Fig. 11B). The external cortex is then formed by a thinner layer of poorly vascularized tissue with at least three distinct rest lines, one of which essentially marks the surface of the femur. In ROMVP 81200, there appear to be two zones of growth, one adjacent to the medullary cavity and one along the external cortex, and one annulus with at least three continuous rest lines (Fig. 12B). The second growth zone is not fully continuous throughout, but the annulus can be traced circumferentially such that it can be determined to precede this zone. The observation that the most peripheral zone in ROMVP 80916 is an annulus, whereas the corresponding zone in ROMVP 81200 is a growth zone, indicates that the animals died during different seasons. For both specimens, a minimum age of two years can thus be estimated.

## DISCUSSION

**Intraspecific variation.** Described specimens of *Seymouria* illustrate the intraspecific variation that occurs across the postcranial skeleton. For example, the porosity of the diapophyses differs between each vertebra in OMNH 79348, suggesting differential stress loading on each vertebra. The identity of the vertebra immediately posterior to the first sacral has also been historically debated; *White (1939)* and *Berman et al. (2000)* identified the element as a second sacral, while *Williston (1911)*, *Watson (1918)*, and *Berman, Reisz & Eberth (1987)* identified it as the first caudal. *White*'s (*1939*) argument was based on the presence of a rib extending anteriorly from the vertebra in question to contact the rib of the first sacral, stating unequivocally that it must have had a strong cartilaginous connection and was doubtlessly a functional second sacral vertebra (p. 354). In contrast, *Berman, Reisz & Eberth (1987)* noted that the rib of the debated element curved posteriorly and thus could not contact the ilium or the rib of the first sacral, negating any possibility of it forming a true second sacral. In the absence of a preserved rib in OMNH 79348, we are unable to comment on whether the vertebra immediately following the first sacral represents a definitive second sacral or the first caudal vertebra.

Compared to the smaller and presumably more immature humerus (FMNH PR 2054) that was previously described from Richards Spur (*Sullivan & Reisz, 1999*: Fig. 2), many features on OMNH 74721 are more developed. In FMNH PR 2054, the capitellum is

indicated only by a slight swelling, whereas in OMNH 74721 it is a distinct hemispherical facet. Mirroring this disparity, the supinator process of FMNH PR 2054 remains connected to the capitellum by a ridge of unfinished bone, whereas the process of OMNH 74721 is clearly delineated from the distal articular surfaces by an emargination of smoothly finished bone.

**Comparative external anatomy.** Comparisons of *Seymouria* with many other seymouriamorphs, especially with the discosauriscids, are complicated by the absence of definitively adult individuals of many seymouriamorph taxa. Further complications arise from the evident taphonomic distortion and compaction of many seymouriamorph specimens. Any discussion of comparative anatomy among seymouriamorphs, including the comparisons presented here, must be viewed within the context of these limitations.

The vertebrae of *Seymouria* bear a number of marked differences from those of other seymouriamorphs. In *Ariekanerpeton*, the presacral vertebrae differ in the level of ossification of the neural arch, with the arches of the third to fifth vertebrae remaining separate and the sixth to ninth arches being strongly co-ossified (*Klembara, 2005*). The neural spines of *Kotlassia* are much higher than in *Seymouria* (*Bystrow, 1944*), whereas in *Discosauriscus* the spines are short (*Klembara & Bartík, 2000*), and in *Ariekanerpeton* the spines are either low or unossified altogether (*Laurin, 1996b*). The vertebrae of *Utegenia* have been described as indistinguishable from those of *Discosauriscus* (*Klembara & Ruta, 2004*). The oval cross-section of the diapophyses is shared between *Seymouria* (*White, 1939*) and *Kotlassia* (*Bystrow, 1944*). However, in *Seymouria* the diapophyses remain the same size but transition in orientation from vertical to horizontal along the vertebral column (*White, 1939*), whereas the diapophyses of *Kotlassia* maintain the same orientation but decrease in size along the column (*Bystrow, 1944*). Unlike *Seymouria* and *Kotlassia* in which the articular surfaces of the prezygapophyses and the postzygapophyses face directly dorsally and ventrally, respectively (*White, 1939*; *Bystrow, 1944*; *Holmes, 1989*), the surfaces of *Discosauriscus* and *Ariekanerpeton* face dorsomedially and ventrolaterally (*Klembara & Bartík, 2000*; *Klembara, 2005*). *Seymouria*, *Kotlassia*, and *Discosauriscus* retain an open notochordal canal in adulthood (*White, 1939*; *Bystrow, 1944*; *Klembara & Bartík, 2000*). It is unknown whether the notochordal canals were retained in the other seymouriamorphs (*Klembara & Ruta, 2004*; *Klembara, 2005*; *Klembara, 2005*; *Klembara, 2009*; *Bulanov, 2014*). *Discosauriscus* has only one sacral vertebra (*Klembara & Bartík, 2000*), as in OMNH 79348. It is unclear whether *Seymouria baylorensis* and *Seymouria sanjuanensis* differ in the number of sacral vertebrae (*White, 1939*; *Berman, Reisz & Eberth, 1987*; *Berman et al., 2000*), so this character cannot be used to clarify the specific designation of this material.

The humerus of *Seymouria* is more extensively ossified than those of *Utegenia* or *Ariekanerpeton* (*Laurin, 1996a*; *Laurin, 1996b*; *Klembara, 2005*), in which the capitellum, the trochlea, the supinator process, and the deltopectoral crest are indistinct in even the largest individuals. In particular, the humerus of *Utegenia* has been described as "little more than a stout, subcylindrical lump of bone" (*Klembara & Ruta, 2004*: 77). It has been noted, however, that the lack of ossification in *Utegenia* may represent ontogenetic immaturity (*Klembara & Ruta, 2004*). The humerus of *Kotlassia* is the slenderest of the seymouriamorphs and lacks an entepicondylar foramen (*Bystrow, 1944*). *Makowskia*,

*Spinarerpeton*, and *Discosauriscus* exhibit the same massive, L-shaped deltopectoral crest extending from the proximal articular surface as that seen in *Seymouria* (*Klembara, 2005*; *Klembara, 2009*; *Klembara & Bartík, 2000*). The humeri of *Makowskia* and *Spinarerpeton* are described as having a broad shaft, a proximodistally short entepicondyle, and an entepicondylar foramen that is open distally (*Klembara, 2005*; *Klembara, 2009*); the last feature has been recognized as likely being ontogenetic. *Makowskia*, *Spinarerpeton*, and *Discosauriscus* differ substantially from *Seymouria* in having a well-developed insertion site for the *m. subcoracoscapularis* and in lacking a supinator process (*Klembara & Bartík, 2000*; *Klembara, 2005*; *Klembara, 2009*); in *Seymouria*, the former is represented by only a slight rugosity while the latter is quite pronounced (*White, 1939*). The humeri of *Discosauriscus* and *Ariekanerpeton* have been reported with a lesser degree of twisting, with the planes of the proximal and distal ends being at an approximately 38-degree angle (*Klembara & Bartík, 2000*; *Klembara, 2005*). The positioning and degree of development of the insertions for the *m. subcoracoscapularis* and *m. latissimus dorsi* are roughly equivalent in *Discosauriscus* as in *Seymouria* (*Klembara & Bartík, 2000*).

As with the humerus, the femur of *Seymouria* is more ossified than in *Utegenia* or *Ariekanerpeton* (*Laurin, 1996a*; *Laurin, 1996b*); the femur of *Utegenia* has been described as featureless (*Klembara & Ruta, 2004*). The femur of *Makowskia* is known but has only been described as having a crescentic proximal articular surface with a deep intertrochanteric fossa as in *Seymouria* (*Klembara, 2005*), and the only corresponding figure is a simplified line drawing. The femur of *Spinarerpeton* has been similarly described and figured in limited detail (*Klembara, 2009*). The femur of *Microphon* is much more gracile than in *Seymouria* with less pronounced expansion of the epiphyses and a proportionally longer, narrower shaft (*Bulanov, 2014*). *Microphon* also differs from *Seymouria* in that the adductor crest extends towards the fibular condyle (*Bulanov, 2014*), as opposed to the tibial condyle as occurs in *Seymouria* and *Discosauriscus* (*Klembara & Bartík, 2000*). In addition, the intertrochanteric fossa of *Microphon* is approximately one-third the length of the femur (*Bulanov, 2014*), whereas in *Seymouria* it extends approximately halfway along the element. *Kotlassia* appears to be intermediate between these two conditions and exhibits a starker contrast between the slender shaft and the broadly expanded epiphyses (*Bystrow, 1944*: Fig. 9). A feature found in the femora of *Microphon* and *Kotlassia* and in ROMVP 80915, though perhaps not all specimens of *Seymouria*, is the lesser development of the trochanter (*Bystrow, 1944*); in *Microphon*, the trochanter seems to not be developed at all (*Bulanov, 2014*). The femur of *Discosauriscus* is similar to that of *Seymouria* in overall morphology but is more gracile (*Klembara & Bartík, 2000*: Fig. 25).

The fibulae of *Kotlassia* and *Utegenia* have not been illustrated or described in sufficient detail to allow for accurate comparison with *Seymouria*. The fibula of *Ariekanerpeton* has only been described as being poorly preserved with unfinished epiphyses (*Klembara, 2005*). In *Makowskia* and *Discosauriscus*, the fibula is similar to *Seymouria* in being deeply concave medially with only a slight concavity on the lateral surface (*Klembara & Bartík, 2000*; *Klembara, 2005*). The fibula of *Spinarerpeton* is unknown (*Klembara, 2009*).
## Histological interpretations and comparisons

Contextualizing the histological data of the specimens of *Seymouria* is complicated by the paucity of work on other stem amniotes, let alone seymouriamorphs specifically. Limb elements of *Seymouria* have never been histologically analyzed. The only other seymouriamorph femur to be histologically sampled is that of the European *Discosauriscus*, the femur of which is characterized by a parallel-fibered endosteal matrix with sparse vascularization comprised of radially and longitudinally arranged canals in small individuals that shifts to being dominated by radial vasculature in adults (*Sanchez et al., 2008*). *Seymouria* exhibits a lamellar matrix with a plexiform arrangement of the vasculature in the femur, and the degree of vascularization far exceeds that figured for *Discosauriscus*. The increased vascularization is indicative of relatively fast growth at the time of death in the sampled specimens of *Seymouria*, which also suggests relative immaturity. Skeletochronological markers also differ between the taxa. *Discosauriscus* possesses numerous, well-defined and evenly spaced LAGs, whereas *Seymouria* is characterized by distinctive growth zones and annuli bearing numerous closely spaced rest lines but without clear LAGs. Similar variation in skeletochronological markers has been reported in the Late Triassic metoposaurids *Dutuitosaurus* from Morocco and *Metoposaurus* from Poland; this disparity was hypothesized to be the result of differing seasonal activity patterns associated with climatic differences across paleolatitudinal gradients (*Konietzko-Meier & Klein, 2013*). Lastly, the medullary spongiosa is distinctly less developed in *Discosauriscus* (*Sanchez et al., 2008*: Fig. 2); the significance of this is unclear in the absence of additional data. It may relate to overall body size, as extant lissamphibians, relatively small in comparison to *Seymouria*, typically lack medullary spongiosa; however, the larger cryptobranchids possess a spongiosa (e.g., *Laurin, Canoville & Germain, 2011*). Similarly, small, semi-terrestrial to terrestrial temnospondyls also have little to no spongiosa (e.g., *McHugh, 2015*) compared to the larger *Eryops* with a much denser spongiosa (e.g., *Konietzko-Meier, Shelton & Sander, 2016*).

Comparisons with other Paleozoic tetrapods are also limited by a paucity of comparative work. Of the major Paleozoic clades (e.g., pelycosaurian synapsids, 'lepospondyls'), temnospondyls are the best-sampled (*Sanchez et al., 2010a*; *Sanchez et al., 2010b*; *McHugh, 2014*; *Konietzko-Meier, Shelton & Sander, 2016*). The relative thickness of the cortex and the development of the medullary spongiosa are most comparable to that of the co-occurring trematopid *Acheloma dunni*, a terrestrial taxon (*Sanchez et al., 2010b*; *Quemeneur, de Buffrénil & Laurin, 2013*). The spongiosa is less developed than in either definitively aquatic taxa such as the late Permian rhinesuchid *Rhinesuchus* (*McHugh, 2014*) or in controversially aquatic taxa such as the early Permian eryopid *Eryops* (*Sanchez et al., 2010b*; *Quemeneur, de Buffrénil & Laurin, 2013*; *Konietzko-Meier, Shelton & Sander, 2016*), and the cortex is not extensively thickened as in the definitively aquatic dvinosaur *Trimerorhachis* (*Sanchez et al., 2010b*; *Quemeneur, de Buffrénil & Laurin, 2013*). A large number of Mesozoic temnospondyls, which are predominantly aquatic, have also been sampled (*Steyer et al., 2004*; *Konietzko-Meier & Sander, 2013*; *Sanchez & Schoch, 2013*). Many of these taxa exhibit similar structure to that of *Trimerorhachis*, often with a high degree of pachyostotic development and with greatly reduced or nearly absent medullary

cavities. *McHugh (2015)* sampled the small-bodied Early Triassic lydekkerinid *Lydekkerina*, a semi-aquatic taxon with terrestrial capabilities (e.g., *Pawley & Warren, 2005*; *Canoville & Chinsamy, 2015*) and the amphibamiform *Micropholis*, a fully terrestrial taxon (e.g., *Schoch & Rubidge, 2005*). Both taxa exhibit a similar histological and microanatomical organization to that of terrestrial Paleozoic temnospondyls and to that of *Seymouria*. Collectively, the temnospondyl comparisons support an inferred terrestrial lifestyle of *Seymouria*. However, it is important to note that the spongiosa of *Seymouria* is more developed than in any of the co-occurring terrestrial temnospondyls at Richards Spur (*Castanet et al., 2003*; *Quemeneur, de Buffrénil & Laurin, 2013*; *Richards, 2016*; *Gee, Haridy & Reisz, 2020*) in which the spongiosa is either weakly developed (Trematopidae) or virtually non-existent (Dissorophidae, Amphibamiformes). However, it is comparable in the developed medullary spongiosa to that of *Eryops*, the degree of terrestriality of which has long been debated (*Konietzko-Meier, Shelton & Sander, 2016*, and references therein). The significance of the spongiosa in *Seymouria* is uncertain at present and warrants further work to compare with other co-occurring taxa and with more closely related stem amniotes.

The vertebral histology is also difficult to compare with coeval tetrapods, let alone with closely related taxa. Vertebrae are uncommon in histological studies compared to limb elements, and most studies that that have examined the vertebrae of Paleozoic tetrapods have focused on the inter- and pleurocentra (*Konietzko-Meier, Danto & Gadek, 2014*; *Danto et al., 2017*; *Danto et al., 2019*). However, both the centra and the neural arches contribute valuable information regarding the lifestyle of *Seymouria*. Previous workers have often suggested that the neural arch would have been subject to strong biomechanical constraints during locomotion in early tetrapods (*Rockwell, Evans & Pheasant, 1938*; *Olson, 1976*; *Holmes, 1989*). The prominent expansion of the neural arch and the development of the zygapophyses in *Seymouria* lends support to this hypothesis. *Discosauriscus* is the only other seymouriamorph to have its internal vertebral anatomy examined (*Danto et al., 2016*). The main difference is in the construction of the neural arch, which is comprised of thick, compact lamellar bone in *Discosauriscus*; in contrast, the neural arch of *Seymouria* is largely hollow. Based on the size of the sampled *Discosauriscus* material, the individual was likely premetamorphic and still aquatic, which would explain the higher degree of ossification. Whether this might have changed in later stages of ontogeny if or when individuals metamorphosed into a terrestrial adult form remains unknown. Beyond seymouriamorphs, neural arches have not been sampled in many clades, which may be because most Paleozoic tetrapod clades have multipartite vertebrae in which the arch readily detaches from the centra during preservation. Furthermore, isolated neural arches have not traditionally been utilized as an ideal case study for exploring histological questions compared to either the centra or to other postcranial elements. *Danto et al. (2016)* sampled a number of Paleozoic lepospondyl taxa in which neural arches were preserved. Some of the aquatic taxa (e.g., an indeterminate nectridean) exhibit a similar spongy bone composition of the arch, but the interior of the arch is relatively well-ossified with little empty space.

**Skeletochronological interpretations.** In the absence of comparative histological data, most inferences regarding the life history of seymouriamorphs have been based on external anatomy of different skeletal regions. For example, most individuals of *Discosauriscus*

retain lateral line canals on the skull, indicating an aquatic lifestyle, but this may also reflect a biased relative abundance of premetamorphic individuals in the fossil record (e.g., *Klembara et al., 2006*). Although definitive adults of this taxon, terrestrial or otherwise, are unknown (*Klembara, Martens & Bartík, 2001*), previous authors have inferred that *Discosauriscus* underwent metamorphosis (*Klembara, 1995*), or that if some species were paedomorphic, they were derived from an ancestor that did metamorphose into a terrestrial adult (e.g., *Boy & Sues, 2000*). For *Discosauriscus austriacus*, *Sanchez et al. (2008)* reported that metamorphosis occurred around the sixth year of life. Although not explicitly stated as such, determination of metamorphosis in that study was rendered feasible through the sampling of limb material of *Discosauriscus* from articulated skeletons that would permit correlation of the skeletochronological data from the histological analysis with external osteological features traditionally used for relative age determination.

Characterizing ontogeny in *Seymouria* is complicated by a general paucity of associated postcrania compared to *Discosauriscus* such that most characterizations are solely based on the cranium (e.g., *Klembara et al., 2006*; *Klembara et al., 2007*). Indeed, it is not known for certain that *Seymouria* underwent metamorphosis, as no larval forms have been recovered. The smallest reported specimen is one from the Bromacker quarry with a skull measuring 2.1 cm (*Berman & Martens, 1993*), but the preservation is too poor to be described for comparative anatomical purposes. Most of the other well-described cranial specimens exceed 8.0 cm (e.g., *Berman, Reisz & Eberth, 1987*), and small specimens remain poorly represented (e.g., *Klembara et al., 2006*). Simultaneously, the histological signals of metamorphosis in early tetrapods, if they exist, remain poorly understood. In extant lissamphibians, which may be the most appropriate extant analogue for seymouriamorphs, there is often a 'transformation mark' (*Schroeder & Baskett, 1968*) or a 'metamorphosis line' (*Rozenblut & Ogielska, 2005*) that demarcates overwintering following metamorphosis. Unfortunately, the criteria for identifying this line are somewhat ambiguous in the literature beyond it being the closest circumferential line to the medullary cavity and likely being closely spaced to the first LAG. Identifying this feature may thus rely mainly on an absence of endosteal resorption (present in our sectioned specimens) that would obliterate this mark. The line has a similar appearance to LAGs (e.g., *Tsiora & Kyriakopoulou-Sklavounou, 2002*), and there is some debate over whether there is a correlation of this line with deposition of woven-fibered bone (e.g., see *Castanet & Smirina, 1990*; *Guarino et al., 2003*). To the best of our knowledge, this feature has never been identified in a Paleozoic tetrapod, and the applicability of crown lissamphibian life history to seymouriamorphs on the amniote stem is unknown.

As a result, determining whether a specimen of *Seymouria* is 'postmetamorphic' is based largely on an established precedent of using external features from which admittedly arbitrary and gradational terms such as 'juvenile,' 'sub-adult,' and 'adult' are derived. It bears noting that these terms may refer to different biological attributes (e.g., sexual maturation vs. the process of metamorphosis) and may be used differently by various workers. With respect to ROMVP 80916, the large partial femur that was sectioned, there is a well-established precedent for identifying the element as belonging to a probable 'adult' that had completed metamorphosis. ROMVP 80916 is incomplete, but the preserved

portion is the same size as the complete ROMVP 80915, which measures 5.5 cm in length and which is on the larger end of previously reported specimens (e.g., 6.4 cm; *White, 1939*). Based on comparisons with articulated specimens of *Seymouria* (*Berman, Reisz & Eberth, 1987*; *Berman et al., 2000*), this femur would belong to an individual with a skull length exceeding 10 cm, which falls well within a range for which specimens have been previously described as 'adults.' Femora of the previously described articulated skeletons are smaller than ROMVP 80916 yet the skulls possess numerous features accepted as evidence for both somatic maturity and terrestriality, such as the absence of lateral line grooves, ossified carpals and tarsals, firmly interdigitated sutures, and pronounced ornamentation (*Boy & Sues, 2000*). ROMVP 81200 is smaller, measuring only 2.6 cm as preserved. By comparison with ROMVP 80915, assuming isometric scaling of the element, ROMVP 81200 would have been around 4.2 cm in length when preserved. Using this estimate and comparisons with articulated specimens, the skull of this individual would have been around 8 cm, which is only slightly below the lower size bound reported for most specimens of *Seymouria* (e.g., *Berman, Reisz & Eberth, 1987*) and which would at least represent a 'sub-adult' based on previous designations. Further evidence for a postmetamorphic determination may be found in the nature of the preservational environment of Richards Spur. Beyond the enigmatic and extremely rare aïstopod *Sillerpeton permianum*, there is no evidence of aquatic tetrapods, either larval forms of metamorphosing adults or obligately aquatic adult forms, even though material of very small-bodied tetrapods is captured (*MacDougall et al., 2017*). Regardless of whether this represents a biased sample, it is clear that the fissure fills were not conducive to the capture of aquatic tetrapods.

Based on a postmetamorphic interpretation of the sectioned material, the onset of metamorphosis in *Seymouria* may thus be constrained to probably occurring by the second year of life. It may be inferred that the larger ROMVP 80916 was probably older than two years, although this cannot be proven nor can the amount of growth cycles lost to remodeling be determined at present. Metamorphosis in *Seymouria* occurred much earlier than in *Discosauriscus*, which was suggested to undergo a delayed metamorphosis around the sixth year of life (*Sanchez et al., 2008*). The more compact and sparsely vascularized lamellar bone of *Discosauriscus* also support interpretations of slower growth and a protracted aquatic larval stage in this taxon (*Sanchez et al., 2008*). This disparity among closely related taxa may reflect the different environments that these taxa inhabited (correspondent with their differing lifestyles), as *Seymouria* is primarily found in fluvial environments of North America, and *Discosauriscus* is primarily found in lacustrine settings in Europe. Both taxa likely experienced distinct seasonality, but the presence of distinct LAGs in *Discosauriscus* suggests a stronger relative influence of environmental fluctuations on this taxon. Some seasonality at Richards Spur, previously demonstrated through other lines of evidence (e.g., *Woodhead et al, 2010*; *MacDougall et al., 2017*) is evident in our samples from the clearly defined annuli and growth zones (Figs. 11–12), and it seems probable that these environmental conditions (particularly the ephemerality of bodies of water) would have favored an early onset of metamorphosis in *Seymouria*. Taxa living in the same habitat and experiencing similar conditions can exhibit different responses to those conditions—amniotes are typically less constrained by water stress

relative to non-amniotes, for example. A hypothesis of variable responses to similar climate at Richards Spur is supported by the differing skeletochronological records of co-occurring tetrapods from the site. The dissorophoid temnospondyls exhibit clear LAG patterns (*Richards, 2016*; *Gee, Haridy & Reisz, 2020*), whereas the varanopid synapsids exhibit a mixture of LAGs and annuli with growth zones (*Huttenlocker & Shelton, 2020*), and the captorhinid eureptiles exhibit only annuli with growth zones (*Peabody, 1961*; *Richards, 2016*; *Huttenlocker & Shelton, 2020*). Other hypotheses to explain the differing patterns include secondary environmental factors that may themselves be influenced by climate but in more nuanced and asymmetrical fashions, such as prey availability or limited spatial occupation within the habitat.

Future work will be necessary to identify histological markers of pre- versus postmetamorphic life stages. The present study cannot inform further on this because of the absence of definitive larval specimens of *Seymouria*, which in turn creates more uncertainty regarding the nature of metamorphosis, if it occurred in this taxon, as well as the external osteological changes associated with this process if it did. Even if our interpretation of the sampled specimens as postmetamorphic is accepted, the timing of either the onset or completion of metamorphosis relative to the time of death remains unclear. In order to more precisely constrain histological markers of metamorphosis will require sampling of taxa, such as *Discosauriscus*, for which metamorphosis is definitively known and for which osteological changes associated with this process are well-documented.

**Lifestyle interpretations.** In early tetrapods, interpretations of lifestyle (e.g., aquatic vs. terrestrial) are often based on the presence or absence of features such as lateral line grooves and the degree of development of external features of the limbs (e.g., *Moodie, 1908*; *Schoch, 2002*; *Witzmann, 2016*). Histology has more recently been utilized as a means to further test these hypotheses by means of comparisons with extant taxa in which mode of life can be definitively observed and with the classically utilized external anatomical features (*Germain & Laurin, 2005*; *Kriloff et al., 2008*; *Sanchez et al., 2010a*; *Quemeneur, de Buffrénil & Laurin, 2013*; *Konietzko-Meier, Shelton & Sander, 2016*). *Seymouria* is widely accepted to lack lateral line grooves, although they have been suggested by some to have been present in juveniles (*Berman & Martens, 1993*, but see *Klembara et al., 2006*), suggesting a transition in lifestyle throughout ontogeny. Additionally, the limb bones are well-developed, with prominent attachment sites for musculature and distinct processes (Figs. 4–5), and the neural arches are greatly expanded compared to other Paleozoic tetrapods with prominent zygapophyses inferred to have supported the axial column in a terrestrial animal (*Sullivan & Reisz, 1999*). Our data further corroborate the interpretation of the Richards Spur locality as an assemblage dominated by terrestrial fauna (*MacDougall et al., 2017*).

The centra also contribute information through inferences on the skeletal mass of the element(s). The two traditionally utilized criteria are the thickness of the periosteal domain and the presence or absence of calcified cartilage. Greatly thickened domains (pachyostosis) and retention of calcified cartilage throughout ontogeny are frequently seen in large-bodied aquatic temnospondyls and probably served to increase the skeletal mass for buoyancy control (*Danto et al., 2016*). In both *Discosauriscus* and *Seymouria*, the periosteal domain is relatively thin, and calcified cartilage is primarily found around the notochordal canal

(*Danto et al., 2016*). In *Seymouria*, this is the only location of this tissue, whereas calcified cartilage occurs sporadically in the endochondral domain of at least immature individuals of *Discosauriscus*.

What then can be concluded regarding the histological data from *Seymouria* postcrania and the lifestyle of the taxon? The femoral microanatomy, specifically the relatively thin cortex and the modest development of the medullary spongiosa, is more compatible with that of a terrestrial animal by comparison with other Paleozoic tetrapods (primarily temnospondyls) that have been inferred to be terrestrial (e.g., *Sanchez et al., 2010b*; *Quemeneur, de Buffrénil & Laurin, 2013*). Based on studies of femoral and tibial microanatomy in extant tetrapods (*Kriloff et al., 2008*; *Quemeneur, de Buffrénil & Laurin, 2013*), these features also support a primarily terrestrial lifestyle. Collectively, this corroborates the conclusions of previous authors that *Seymouria* was most likely a terrestrial animal (*White, 1939*; *Berman & Martens, 1993*; *Sullivan & Reisz, 1999*; *Marchetti, Mujal & Bernardi, 2017*). The vertebral histology also confers support for a terrestrial lifestyle. The periosteal domain is thin, calcified cartilage is sparse and confined to the margin of the notochordal canal, and the neural arch is largely hollow. These data correspond favorably with the broad expansion of the arch and the zygapophyses, which *Sullivan & Reisz (1999)* interpreted to be for the stiffening of the axial column following *White (1939)*.

**Terrestriality in seymouriamorphs.** Assessing the range of ecologies among seymouriamorphs from a macroevolutionary standpoint is important because the group has historically been regarded as being well-situated for understanding the skeletal modifications associated with terrestriality. *Seymouria* is one of the best seymouriamorphs for examining such modifications because complete, articulated skeletons are known (e.g., *Berman, Reisz & Eberth, 1987*), but it then becomes important to assess whether a terrestrial or aquatic lifestyle is the plesiomorphic state among seymouriamorphs. Given that seymouriamorphs, and reptiliomorphs more broadly, are frequently used as exemplars for the skeletal changes associated with terrestrial adaptation, clarifying the primitive condition of this group is critical for informing accurate comparisons. A conceptual phylogeny, adapted from *Klembara (2011)*, is presented in Fig. 14, illustrating the distribution of lifestyles among seymouriamorphs.

Our data provide strong evidence at the histological and microanatomical scales to support the longstanding hypothesis of terrestriality in *Seymouria*. This is not a particularly controversial idea; numerous aspects of the external morphology, such as the well-ossified limbs and the massively expanded vertebrae have long been cited as evidence for this lifestyle (*Romer, 1956*). Although *Berman & Martens (1993)* described a possible indication of a lateral line system in juvenile specimens of *S. sanjuanensis* from Germany, subsequent work (*Klembara et al., 2006*) on an early juvenile did not find any evidence for a lateral line system in other *S. sanjuanensis* from the same locality. As such, while it is often inferred that *Seymouria* underwent metamorphosis as with other seymouriamorphs and a number of other terrestrial tetrapods (e.g., some temnospondyls), definitive aquatic larval forms and morphological transitions associated with the presumed metamorphosis are unknown.

At least one other seymouriamorph, *Karpinskiosaurus*, is also represented only by specimens that lack lateral line grooves (*Klembara, 2011*). *Kotlassia* has also been historically

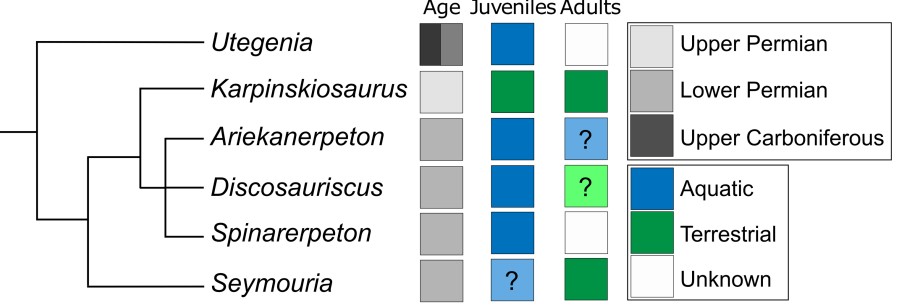

**Figure 14** **Conceptual phylogeny adapted from *Klembara (2011)*, illustrating the distribution of aquatic and terrestrial taxa among seymouriamorphs.** Blue boxes represent taxa interpreted as being aquatic. Green boxes represent taxa interpreted as terrestrial. Grey boxes represent taxa for which no interpretation has been made. Boxes in pale blue or green containing question marks indicate taxa for which there have been some suggestions regarding the possible ecology of a given ontogenetic stage, but for which there are no known specimens that definitively verify the suggested interpretations.

regarded as lacking lateral line grooves (e.g., *Bystrow, 1944*), but the *Kotlassia* of most previous authors is actually a combination of material referable to the type species, *Kotlassia prima*, and material properly referable to *Karpinskiosaurus* (see *Bulanov, 2002*; *Klembara, 2011*, for discussion). Whether these grooves are definitively absent in the holotype of *Ko. prima* is not apparent from previous works that accounted for this historical discrepancy. For *Karpinskiosaurus* and *Seymouria*, it has been proposed that these taxa underwent metamorphosis relatively early in their development and lived on land for the majority of their lives (*Klembara, 2011*).

In contrast, most other seymouriamorphs are known from individuals with lateral line grooves, including *Ariekanerpeton* (*Klembara, 2005*), *Discosauriscus* (*Klembara, 1996*), *Spinarerpeton* (*Klembara, 2009*), and *Utegenia* (*Malakhov, 2000*). The most recent phylogenetic analysis that focused on seymouriamorph phylogeny is that of *Klembara (2011)*, which followed a series of anatomical work that re-described virtually all known seymouriamorphs. Mapping the distribution of ecologies onto this topology suggests that seymouriamorphs are primitively aquatic (*Utegenia* being the earliest diverging taxon) with two separate shifts to terrestriality, one in *Karpinskiosaurus* and one in *Seymouria* (Fig. 14). However, caution must be exercised in inferring the phylogeny of a clade in which metamorphosis is known to occur, because biases in the record of premetamorphic larval forms versus that of postmetamorphic terrestrial adults (if such determinations can be made to begin with) can produce misleading data. As with *Seymouria* (*Berman et al., 2000*), it has been proposed that *Discosauriscus* transitioned from an aquatic to terrestrial lifestyle throughout its ontogeny, but even the largest known specimens of *Discosauriscus* are believed to be juveniles, and none have been recovered from the terrestrial environments that the adult individuals may have inhabited (*Klembara, Martens & Bartík, 2001*). This may relate to a relatively protracted larval stage that was recovered through the histological work of *Sanchez et al. (2008)* in which metamorphosis may not have begun until year six of an individual's life. The latest phylogenetic analyses (*Klembara, 2011*) do not bear

out the slippage that is predicted when coding taxa based on immature specimens (i.e., *Discosauriscus* is a highly nested seymouriamorph), but this does not negate the potential for this disparity to affect the phylogeny. *Ariekanerpeton*, *Spinarerpeton*, and *Utegenia* are also likely represented only by juveniles (*Klembara & Ruta, 2003*; *Klembara & Ruta, 2005*; *Klembara, 2009*), which warrants consideration.

## CONCLUSIONS

Histology offers one avenue for exploring the diversity of life histories within a clade through well-documented taxa (e.g., *Discosauriscus, Seymouria*) and for improving hypotheses and predictions regarding those of more poorly represented taxa with limited ontogenetic data. The correlation of the skeletochronological data from our histological analysis with the relative size and development of external features of the sampled femora substantiate the hypothesis that *Seymouria* was a rapidly metamorphosing taxon that spent most of its life on land (*Klembara, 2011*). This life history may explain why it is predominantly found in fluvial deposits of south-southwestern North America and the upland Bromacker locality along with other highly terrestrial tetrapods whereas the slower growing *Discosauriscus* is restricted to lacustrine environments of Europe. A relatively early onset of metamorphosis may also account for the absence of larval forms of *Seymouria* (i.e., a short aquatic larval stage) and the probable spatial separation of larvae from the fluvial environments that preserved the adults (niche partitioning between life stages). Recovery of small and presumably immature specimens of *Seymouria* will be required to further explore the taxon's life history and to contextualize it with other terrestrial tetrapods.

In the sense that terrestriality in adults of *Seymouria* has not been widely questioned, our most novel data, the histological data, are not necessarily surprising. However, this should not diminish the value of these data; testing hypotheses using multiple approaches is important for assessing the rigor of such hypotheses. The existing histological framework and understanding of seymouriamorph development remains largely confined in traditional interpretations of relative maturity based on external features and their development, and there is an extensive precedent for the utility of histology (among other more recently accessible methods) to further explore paleobiological attributes of extinct taxa. Our interpretations of the data are somewhat limited, in part by sample size, but also in part by the absence of a substantive body of comparative data. It is unclear, for example, what to make of the persistence of a modestly developed medullary spongiosa in the femur of *Seymouria*, either compared to *Discosauriscus* or to other more distantly related terrestrial tetrapods (though see our previous comment regarding the potential for there to be a size correlation). Taken in isolation, the presence of this spongiosa could indicate that *Seymouria* was more semi-terrestrial than previously believed, at least at the captured stages of its life history, but it bears reiterating that microanatomy captures more than just a signal of ecological lifestyle, such as signals from phylogeny or from life history. Lifestyle is also gradational, both between and within taxa, such that binning taxa into discrete categories can prove challenging. Our ontogenetic trends are more accurately stated as two partial points within the developmental trajectory, and the opportunity remains to explore ontogeny further within *Seymouria* across all regions of the skeleton.

**Institutional Abbreviations**

**FMNH**     Field Museum of Natural History, Chicago, IL, USA
**OMNH**     Sam Noble Oklahoma Museum of Natural History, Norman, OK, USA
**ROMVP**    Royal Ontario Museum, Toronto, ON, Canada

## ACKNOWLEDGEMENTS

Thanks to Kevin Seymour (ROMVP) and Jennifer Larsen (OMNH) for assistance with collection numbers. Thanks to Ashley Reynolds and Jade Simon (ROMVP) for assisting KDB in the production of histological thin sections. Thanks to Yara Haridy (Museum für Naturkunde Berlin) for discussion of histology. Thanks to Sophie Sanchez (Uppsala University) for sharing thin section images of *Discosauriscus*. Thanks to Michel Laurin (Muséum National d'Histoire Naturelle) for discussion of limb microanatomy.

### Funding

This work was supported by grants from the University of Toronto and the Natural Sciences and Engineering Research Council of Canada (NSERC) to Robert R. Reisz, an NSERC Scholarship to Kayla D. Bazzana, and an Ontario Graduate Scholarship (OGS) to Bryan M. Gee. The funders had no role in study design, data collection and analysis, decision to publish, or preparation of the manuscript.

### Grant Disclosures

The following grant information was disclosed by the authors:
University of Toronto.
Natural Sciences and Engineering Research Council of Canada (NSERC).
NSERC Scholarship.
Ontario Graduate Scholarship.

### Competing Interests

The authors declare there are no competing interests.

### Author Contributions

- Kayla D. Bazzana and Bryan M. Gee conceived and designed the experiments, performed the experiments, analyzed the data, prepared figures and/or tables, authored or reviewed drafts of the paper, and approved the final draft.
- Joseph J. Bevitt performed the experiments, authored or reviewed drafts of the paper, and approved the final draft.
- Robert R. Reisz conceived and designed the experiments, authored or reviewed drafts of the paper, and approved the final draft.

## Data Availability

Unprocessed 16-bit TIFF slices are available at MorphoBank: https://morphobank.org/index.php/Projects/ProjectOverview/project_id/3549 (*O'Leary & Kaufman, 2012*).

Specimens OMNH 74721 and OMNH 79346-79353 are held by the Sam Noble Oklahoma Museum of Natural History in Norman, Oklahoma, USA. Specimens ROMVP 80915-80917 and ROMVP 81198-81200 are held by the Royal Ontario Museum in Toronto, Ontario, Canada.

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
