# Peer review of "Postcranial anatomy and histology of Seymouria, and the terrestriality of seymouriamorphs"

_PeerJ, doi:10.7717/peerj.8698_

## Round 0.1 · original submission · Minor Revisions

Apologies with some delay as we were awaiting a 3rd review but they have taken too long, so we are proceeding. Thankfully the 2 reviewers broadly agree that the paper is interesting and sound, but have some recommendations for moderate revisions- mainly reconsidering conclusions about postmetamorphic stage and about terrestriality. We may or may not require re-review-- please ensure your Response to Reviewers covers all points raised. Thank you.

·

Basic reporting

Dear Editor of PeerJ,
Dear Authors,

I have reviewed the manuscript entitled Postcranial anatomy and histology of Seymouria, and the terrestriality of seymouriamorphs by Kayla D Bazzana, Bryan M Gee, Joseph J Bevitt and Robert R Reisz.

It is a very interesting study about histology and morphology of Seymouria and implications for the interpretation of the mode of life and it was a pleasure to be able to review this paper. Authors based on the long bones morphology and thin sections proofed that Seymouria was a terrestrial animals. The most important part of this study is the histological description; they are the first known thin sections from long bones from this taxon and even wider, from seymouriamorphs. The only second known histologically seymouriamorphs is Discosauriscus from Europe. Moreover based on the histological sections authors concluded about ontogenetic stage/age of tested bones.
Although the issue is really interesting and deserves to be published in a high ranked journal as PeerJ, I found some problems that must be assessed before final acceptation of the paper. Some of them I have marked directly on the manuscript, general comments you can find below.


Authors used professional English language throughout. My only concern is about saying ecology of Seymouria. I think that ecology it is a more complex term and you are trying the discover/proof mode of life. Of course mode of life (adaptation to environment) is a part of ecology, but I won’t use this as a basic form. To describe ecology you should analyze much more factors.
Authors cited all important papers which are published about this topic. Only small error occurred in MS with the year. Konietzko-Meier and Sander it is 2013 (not 2012).
The structure is mostly clear and conforms to PeerJ standards and discipline norm. The only point is a construction of results. I prefer a separation between results and discussion. In your morphological description you use a lot of external sources. It is acceptable if you are doing a taxonomical paper and need a direct “morphological comparison” with other taxa (taxon A has this processus small, and taxon B has it long, etc). Thus I can understand the mixing of results and discussion in your morphological report. However, still some of your sentences are clear interpretations, and should be in discussion (see pdf).
I would propose to extract histology as a separate part. Also you should add more details about histology. It is a first published description about bone histology of Seymouria and I think it is a good opportunity to provide as detailed as possible description (tissue type, other growth markers, osteocytes, vascularization type, etc.). Especially, if histology is a crucial factor for your conclusions about mode of life and metamorphosis. Moreover, you should add more information about microanatomy, as this is a correct tool to discuss mode of life.
Figures need some improvement. You need more labeling to make all characters described in MS also to be visible on figures. Also in Fig, 4 something happened with the letters, you have no D. For all figures – there is no explanations of used abbreviations.

Experimental design

It is an original primary research within Scope of the journal. Goals are well defined. Methods described with sufficient detail. Beside the morphological description and classical destructive thin sections authors also used modern technologies to access microstructural information – it is an important point, as histology itself is usually a destructive method. Thus all alternatively methods which are safe for fossils are very welcome. The lacking information is about construction of the phylogeny presented on the end of discussion. Authors should add details about this to methods part.

Validity of the findings

For sure it is a very important paper and should be published after improved.

Additional comments

My most important concern is the conclusion about postmetamorphic stage of sampled animals. I agree that this material looks adult-like. But do we know how should look a premetamorphic histology and morphology? I think that it will be useful to add a chapter where you can discuss which characters, morphological and histological, tell us about ontogenetic stage. First, based on literature, then compare data from literature with your material. Only one sentence: “In contrast, the individuals of Seymouria that were analyzed here are regarded as being definitively postmetamorphic in spite of at least one (ROMVP 81200) being distinctly younger than the onset of metamorphosis in Discosauriscus (Sanchez et al., 2008).” is not enough to proof your point. You need provide arguments, both morphological and histological. Now it is difficult to follow your point for researches not familiar with seymouriomorph.
Similar comment about your conclusion about terrestrial mode of life. You need a list of arguments pro and contra, based on literature and your material.

For other comments see pdf.
If you have any further questions please feel free to contact me.
Best regards
Dorota Konietzko-Meier

Reviewer 2 ·

Basic reporting

'no comment'

Experimental design

'no comment'

Validity of the findings

'no comment'

Additional comments

Manuscript “Postcranial anatomy and histology of Seymouria, and the terrestriality of seymouriamorphs” by Kayla D. Bazzana et al. describes the postcranium of the seymouriamorph stem-amniote Seymouria based on outer morphology, bone histology and neutron computed tomography. The postcranium of this early tetrapod has been neglected in comparison to its cranial anatomy, the only comprehensive description of postcranial elements dating back to the first half of the 20th century (White 1939), a description which is very detailed, but which does not meet modern standards anymore. Thus, the description of the postcranium based on new material with new characters is highly welcome. Furthermore, the present manuscript provides data on bone histology and microstructure of limb bones and vertebrae for the first time in Seymouria; these data are essential to assess the mode of life of a fossil tetrapod. The morphological and histological description of this work is always detailed and thorough, and I appreciate the clear and informative illustrations. I also appreciate the documentation of the ontogeny of long bones (humerus and femur), and to my knowledge, this has so far not been described in the literature. The conclusions of the authors are sound and comprehensive, and I agree with most of them. The authors have nicely shown that the larval period must have been rather short in Seymouria in comparison with its relatives (the discosauriscid seymouriamorphs), which had an extended larval period. The greatest surprise for me was to see that the long bones of Seymouria have a modestly developed medullary spongiosa, normally a feature of more aquatic or semi-terrestrial tetrapods. Thus, I suggest that the authors should rethink the hypothesis that Seymouria was strictly terrestrial.
I have some specific suggestions that are listed below. The manuscript is an important step in our understanding of seymouriamorph paleobiology (and thus of stem-amniotes in general), and I fully support its publication in PeerJ.

General remark: I did not find a list of the anatomical abbreviations that are used in the figures.

Specific comments for the authors

Line 38: “Seymouriamorphs are among the best-known stem amniotes…” Although most authors (and I) agree that seymouriamorphs nest on the amniote stem, there is still a different phylogenetic hypothesis (Marjanović & Laurin 2019), stating that seymouriamorphs are stem-tetrapods and might even be paraphyletic (Reference: Marjanović D, Laurin M. 2019. Phylogeny of Paleozoic limbed vertebrates reassessed through revision and expansion of the largest published relevant data matrix. PeerJ 6:e5565 DOI 10.7717/peerj.5565). This unorthodox view should be cited.

Lines 40-44: you should also cite White 1939 here

Lines 121-123: Why do you use Linnean categories like order and family in the Systematic Paleontology section? I would delete them.

Lines 136-143: Description of vertebrae. How do you know or infer the position of vertebrae, even when the column is not complete? Please explain.

Lines 144-145: You cite White 1939 because of the characteristic swollen zygapophyses; I would also cite Holmes 1989 (already cited in the manuscript elsewhere) here that deals with seymouriamorph vertebral morphology and functional aspects

Line 175: supraneural canal: is this a true canal or only an anterior and posterior depression?

Line 176: “A pair of closely spaced, distinctive growth lines can be identified…” Please label in Figure 4A

Line 251: “…cortex is relatively compact…” and well vascularized

Line 252: “less organized”: do you mean woven?

Lines 295-296: you should cite also Holmes 1989 here, and you may mention that also Kotlassia has pre- and postzygapophyses that face dorsally and ventrally, respectively (Bystrow 1944)

Lines 305-329: In this section, you compare the morphology of humerus and femur of Seymouria with those of different discosauriscids. For example, you write that the humerus of Seymouria is more extensively ossified than those of Utegenia or Ariekanerpeton; you state that the lack of ossification may represent ontogenetic immaturity, and I think that is true. We do not know the humeral or femoral morphology in a “true adult” discosauriscid, and therefore such a 1:1 comparison is a bit problematic. You should emphasize this more in the text. Likewise, the lack of a supinator process (line 319) might be an ontogenetic character. The lesser degree of twisting of the humeri of Discosauriscus and Ariekanerpeton (lines 321-323) could also be explained by taphonomy (compaction of the poorly ossified limb)

Line 355: For interpretation of the ecology (aquatic versus terrestrial) you cite Moodie 1908; you should also cite one or more more recent publications. For example, see discussion of this topic in Schoch (2002) and Witzmann (2016) (References: Schoch, R. R. (2002). The evolution of metamorphosis in temnospondyls. Lethaia, 35(4), 309-327; Witzmann, F. (2016). CO 2‐metabolism in early tetrapods revisited: inferences from osteological correlates of gills, skin and lung ventilation in the fossil record. Lethaia, 49(4), 492-506.)

Lines 360-361: Lateral line sulci are indeed absent in Seymouria, however, the absence of the sulci are not definitive proof that Seymouria did not possess a lateral line system. You should argue more cautiously here. The neuromasts could have simply been located more superficially in the (thicker) skin, so that no imprints were made on the bones (e.g. as in lissamphibians, which lack any imprints, but many retain lateral lines). So the presence of grooves are a proof for the presence of the lateral line system, but the absence of sulci is not necessarily a proof for the absence of the lateral line system.

Line 375: “Previous authors have inferred that Discosauriscus underwent metamorphosis” Discosauriscids (or some of them) might have been paedomorphic, but it can be assumed that they were derived from metamorphosing ancestors with terrestrial adults (see e.g. Boy and Sues 2000). Reference: Boy, J. A., and H.-D. Sues. 2000. Branchiosaurs: larvae, metamorphosis and heterochrony in temnospondyls and seymouriamorphs; pp. 1150–1197 in H. Heatwole and R. L. Carroll (eds.), Amphibian Biology, Volume 4: Palaeontology. Surrey Beatty and Sons, Chipping Norton, New South Wales, Australia.

Lines 381-382: “In contrast, the individuals of Seymouria that were analyzed here are regarded as being definitively postmetamorphic in spite of at least one…being distinctly younger…” This is a very interesting observation, please justify why you think they are postmetamorphic.

Line 387: What do you mean with “environmentally stable”? Please be more specific here.

Lines 387-389: “Histological sections also indicate a less developed trabecular network in the medullary cavity of Discosauriscus (Sanchez et al., 2008:fig. 2); the significance of this is unclear in the absence of additional data” This is indeed an interesting contrast to the pattern found in Seymouria. Maybe the less developed trabeculae in Discosauriscus just reflect that the bones are very small. The long bone microstructure in small forms like most lissamphibians is rather simple, with few or no trabeculae, in contrast to larger animals (for example, only cryptobranchids, which are the largest salamanders, possess spongy bone in the tibia). For a discussion see Laurin et al. 2011 and references therein (Reference: Laurin, M., Canoville, A., & Germain, D. (2011). Bone microanatomy and lifestyle: a descriptive approach. Comptes Rendus Palevol, 10(5-6), 381-402.)

Lines 405-406: In my opinion, it is far from certain that Lydekkerina was a terrestrial animal. It has persistent lateral line sulci on the skull roof and Canoville & Chinsamy (2015) – on the basis of bone microstructure – regarded it as “amphibious with a tendency to be more terrestrial” (Reference: Canoville, A., & Chinsamy, A. (2015). Bone Microstructure of the Stereospondyl Lydekkerina Huxleyi Reveals Adaptive Strategies to the Harsh Post Permian‐Extinction Environment. The Anatomical Record, 298(7), 1237-1254.)

Lines 539-541: “…what to make of the persistence of a modestly developed medullary spongiosa in the femur of Seymouria, either compared to Discosauriscus or to other more distantly related terrestrial tetrapod…” As stated above, the rarity of trabeculae in Discosauriscus might be a matter of size of the animal. Concerning the modestly developed medullary spongiosa in Seymouria: maybe this indicates that Seymouria was a bit more aquatic than previously thought? Maybe terrestrial to semi-terrestrial?

Figure 4: “C” on top right of the figure should be D, and the close up picture (E) should be reoriented (rotated through 180°) to be consistent with the overview figure D (“C”)

---

## Round 0.2 · Minor Revisions

A final review has been submitted; the other reviewer was unavailable and I felt a quick check was necessary by at least 1 reviewer. The recommended changes are mainly presentational but there are some scientific critiques as well. Please detail your revisions in a Rebuttal document so I can easily check them and hopefully then would be able to accept the MS. Thank you!

·

Basic reporting

Dear All,

After the second check of submitted article I am happy to see the improvement of the text.
However, I still found some small problems that must be assessed before final acceptation of the paper, comments you can find below, in general comments.

Experimental design

no comments

Validity of the findings

For sure it is a very important paper and should be published after improved.

Additional comments

I think that a little change of the organization of the chapter Material and Methods would make following of the paper easier.
Page 8, line 77: ROMVP 80915 and ROMVP 8091…
Page 9, line 96: For OMNH 79348,….
The list of the material is in a next chapter, so in this moment readers do not know what the number does mean. Maybe add what kind of bone it is (like you did in histological description).

However, in my opinion it will be good to start with the presentation of the material. No you have:
Page 12, lines 157-167 – long “technical” description of vertebrae. Move this better to the part material and methods. And make from this a first part of the section M and M.
Page 14/ line 203 - ROMVP 80915 is a complete left femur (Fig. 5)-… also to M and M
Page 15/68 – fibula – comment see above
Page 17/267-272 – should be in M and M
Etc.

This way you will have a complete list of material, with all necessary information about kind of bone, state of preservation, etc. Now, reading i.e. a discussion, if you have only a coll. number – it is difficult to follow. You have: Referred specimen, but it is only a list. It will be good to have all information together.

Figure caption – fig. 12. You have gz, growth zone - It is enough to say zone. Zone it is always a growing “layer”.
The same figure – you have resting lines. In text you use rest lines – both are correct, but be consistent.

16/253: In both specimens, the neural arch is much thicker along the posterodorsal surface
behind the neural spine and extending down to the posterior indentation of the supraneural canal when compared to the anterior surface. – what do you mean? Cortex is thicker? Or you mean in morphological sense?
16/258: thickened region of ROMVP 81199 (Fig. 8A)… thickened region of cortex?
19/314: For example, the lesser porosity of the diapophyses of the first sacral vertebra relative to the presacral and possible second sacral suggests that the first sacral undergoes the majority of the stress loading.
1. references (usually the lower porosity means lower stress level). 2. Moreover, this sentence somehow does not fit here.
23/422-426: Skeletochronological markers also differ between the taxa. Discosauriscus possesses numerous, well-defined and evenly spaced LAGs, whereas Seymouria is characterized by distinctive growth zones and annuli bearing numerous closely spaced rest lines but without clear LAGs. Lastly, the medullary spongiosa is distinctly less developed in Discosauriscus (Sanchez et al., 2008:fig. 2); the significance of this is unclear in the absence of additional data.
You can observe similar situation for Temnospondyli. Metoposaurus shows thick annuli with numerous resting lines, whereas Dutuitosaurus (metoposaurus from Morocco) normal LAGs. For discussion see: Konietzko-Meier and Klein, 2013.
30-31/560-566: This disparity among closely related taxa may reflect the different environments that these taxa inhabited, as Seymouria is primarily found in fluvial environments of North America that probably experienced marked seasonality, and Discosauriscus is primarily found in lacustrine settings in Europe that may have experienced less seasonal disparity. Such seasonality is evident from the clearly defined annuli and growth zones in our sampled specimens (Figs. 11-12), and it seems probable that these environmental conditions would have favored an early onset of metamorphosis in the taxon.
I would say opposite. Distinct LAGs suggest strong seasonality. See Konietzko-Meier and Klein, 2013.

If you have any further questions please feel free to contact me.
Best regards

---

## Round 0.3 · accepted · Accept

Well done on the revisions. I am satisfied that the reviewer's comments have been adequately addressed and there is no need for further revision or review. Congratulations indeed!